

# An assessment of aerosol optical properties from remote sensing observations and regional chemistry-climate coupled models over Europe

Laura Palacios-Peña[1], Rocío Baró[1,2], Alexander Baklanov[3], Alessandra Balzarini[4], Dominik Brunner[5], Renate Forkel[6], Marcus Hirtl[2], Luka Honzak[7], José María López-Romero[1], Juan Luis Pérez[8], Guido Pirovano[4], Roberto San José[8], Wolfram Schröder[9], Johannes Werhahn[6], Ralf Wolke[9], Rahela Zabkar[10], and Pedro Jiménez-Guerrero[1]

[1]Department of Physics, Regional Campus of International Excellence Campus Mare Nostrum, University of Murcia, Murcia, Spain
[2]Section Chemical Weather Forecasts, Division Data/Methods/Modeling, ZAMG - Zentralanstalt für Meteorologie und Geodynamik, Austria
[3]World Meteorological Organization, Geneve, Switzerland
[4]Ricerca sul Sistema Energetico (RSE), Italy
[5]Laboratory for Air Pollution/Environmental Technology, Empa, Swiss Federal Laboratories for Materials Science and Technology, Switzerland
[6]Karlsruher Institut für Technologie (KIT), Institut für Meteorologie und Klimaforschung, Atmosphärische Umweltforschung (IMK-IFU), Germany
[7]BO-MO d.o.o, Slovenia
[8]Environmental Software and Modelling Group, Computer Science School - Technical University of Madrid, Spain
[9]Leibniz Institute for Tropospheric Research, Permoserstr, Germany
[10]Slovenian Environment Agency, Slovenia

*Correspondence to:* Pedro Jiménez-Guerrero (pedro.jimenezguerrero@um.es)

**Abstract.** Atmospheric aerosols modify the radiative budget of the Earth due to their optical, microphysical and chemical properties, and are considered one of the most uncertain forcing agents. In order to characterise the uncertainties associated with satellite and modelling approaches to represent aerosol optical properties, their representation by different remote sensing sensors and regional online coupled chemistry-climate models over Europe is evaluated. This work also characterises whether

5     the inclusion of aerosol-radiation (ARI) or aerosol-cloud interactions (ACI) helps improve the skills of modelling outputs.

Two case studies were selected within the EuMetChem COST Action ES1004 framework, when important aerosol episodes in 2010 all over Europe took place: a Russian wildfire episode and a Saharan desert dust outbreak that covered most of Mediterranean Sea. The model data came from different regional air quality-climate simulations performed by working group 2 of EuMetChem, which differed according to whether ARI or ACI was included or not. The remote sensing data came from

10     three different sensors: MODIS, OMI and SeaWIFS. The evaluation used classical statistical metrics to first compare satellite data versus the ground-based instrument network (AERONET) and to then model versus the observational data (both satellite and ground-based). The evaluated variables were aerosol optical depth (AOD) and the Angström exponent (AE) at different wavelengths.





Regarding the uncertainty in the satellite representation of AOD, MODIS presented the best agreement with the AERONET observations compared to other satellite AOD observations. The differences found between remote sensing sensors highlighted the uncertainty in the observations, which have to be taken into account when evaluating models. When modelling results were considered, a common trend for underestimating high AOD levels was observed. For the AE, models tended to underestimate
its variability, except when considering a sectional approach in the aerosol representation in the models. The modelling results showed better skills when ARI+ACI interactions were included; hence this improvement in the representation of AOD (above 30% in the model error) and AE (between 20 and 75%) is important to provide a better description of aerosol-radiation-cloud interactions in regional climate models.

## 1 Introduction

The uncertainty of atmospheric aerosol effects on the Earth radiative budget is much greater than for any other climate-forcing agent (Boucher et al., 2013). In this sense, aerosol properties, with emphasis placed on optical properties, are vastly variable on space and time scales due to short-lived aerosol particles and non-uniform emissions (Ramanathan et al., 2001; Kaufman et al., 2002; Randall et al., 2007). Aerosol effects, which depend on their properties, can be classified (according to the IPCC AR5) into Aerosol-Radiation and Aerosol-Cloud Interactions (respectively ARI and ACI) (Boucher et al., 2013).

In order to reduce, or to at least characterise, this uncertainty, there are two main approaches to study aerosol optical properties and their influences on the climate system: (1) measuring observations and (2) using climate models. Aerosol properties cannot be constrained by measurements alone and using models can improve knowledge and our understanding of physical, chemical and optical aerosol properties. However, models are based on the previous observational study of these properties in order to define and implement the behaviour of aerosol particles in modelling systems.

When aerosol optical properties are evaluated, remote sensing is one of the most widely used techniques in the observational approach. The main advantages of remote sensing are: (1) they do not perturb the observed sample (aerosol particles in this case), and are sensitive to different properties (particularly to aerosol optical properties, on which this study focuses); (2) they can provide point, column or profile data. For these reasons, several studies using this approach were carried out to improve knowledge of aerosol properties (e.g., Dubovik et al., 2000; Tanré et al., 2001; Dubovik et al., 2002; Levy et al., 2007; Garland
et al., 2008) and to establish their radiative effects (Haywood et al., 2001; Chou et al., 2002; Bellouin et al., 2005; Huang et al., 2006, among many others).

These studies used data that were measured chiefly from two main platforms: from the Earth?s surface, the so-called ground-based measurement; and from space by using satellites. Moreover, different instruments, such as sun photometers, spectroradiometers or Lidar were used. Major efforts have been made to create networks of ground-based measurements of aerosol
optical properties around the world; e.g., as the Aerosol Robotic Network (AERONET) (Holben et al., 1998), the European Aerosol Research Lidar Network to Establish an Aerosol Climatology (Pappalardo et al., 2014, EARLINET) or the Latin American Lidar Network (Antuña-Marrero et al., 2016, LALINET). There are instruments with onboard satellites that provide information about aerosol optical properties with a wide spatial coverage; e.g., the Multi-angle Imaging SpectroRadiometer



(Diner et al., 1998, MISR), the Moderate Resolution Imaging Spectroradiometer (Remer et al., 2005, MODIS), the Spinning Enhanced Visible and Infrared Imager (Aminou et al., 1997, 1999, SEVIRI), the Ozone Monitoring Instrument (Schoeberl et al., 2006, OMI) or the Cloud-Aerosol Lidar and Infrared Pathfinder Satellite Observation (Winker et al., 2003, CALIPSO).

At this point, it is important to highlight that several studies (Fuzzi et al., 2015; Van Donkelaar et al., 2015) have discussed
the differences between aerosol properties measured by ground-based stations, with a limited spatial coverage, but with more reliable information, and by satellite, with broader spatial coverage, but not with such reliable information as ground-based observations. These aspects are important, together with the wide variability of aerosol optical properties on space and time scales (Ramanathan et al., 2001; Kaufman et al., 2002; Randall et al., 2007), the huge number of different instruments of onboard satellites and the algorithms used, as they produce considerable uncertainty in the measured data.

By considering the modelling approach to reduce or characterise the uncertainty of aerosol effects as part of the COST Action ES1004 EuMetChem (a European framework for online integrated air quality and meteorology modelling; see http://www.eumetchem.info/), a list of interactions were identified as the most relevant coupling processes for regional air quality and weather predictions. These interactions were chosen because experts considered them to be the most important, but were poorly represented at the same time in the current online coupled models (Kong et al., 2015). This type of model offers the
possibility to account for the feedback mechanisms between simulated aerosol concentrations and meteorological variables, and thus permits the simulation of ARI and ACI, which form part of the above-mentioned list of interactions.

The efforts made by EuMetChem action focused on two case studies, which were chosen from the previous experience of phase 2 of the Air Quality Model Evaluation International Initiative (AQMEII Galmarini et al., 2015), when important aerosol episodes took place in 2010 all over Europe with potential aerosol effects on meteorology. These cases consist in the Russian
wildfires and heat wave episode (in July and August 2010) and a Sahara desert dust outbreak with enhanced cloud and rain over the Mediterranean Sea (in October 2010). These case studies were chosen given the evidence of the particularly significant interactions between meteorology and chemistry during such strong air pollution episodes (Konovalov et al., 2011; Chen et al., 2014; Wong et al., 2012).

Several studies have highlighted the relevance of including aerosol radiative feedbacks during such episodes to improve
meteorological forecasts (among many others Pérez et al., 2006; Bangert et al., 2012; Makar et al., 2015b, a). For example, the impacts of high aerosol loading during the desert dust events have been described by Shao et al. (2011) to demonstrate; interactions with climate and ecosystem, impacts on the Earth radiative budget and a drop in photolysis rates.

Furthermore, Konovalov et al. (2011) and Chubarova et al. (2012) have found that the very high aerosol concentration due to the fire emissions during the specific Russian wildfires episode significantly changed the regional conditions of the climate
system by changes in atmospheric gas composition, as well as the optical and radiative characteristics of aerosols. In Péré et al. (2014), the WRF model (Skamarock et al., 2001) was off-line coupled with CHIMERE (Bessagnet et al., 2004; Menut et al., 2014) to study the ARI effects during the Russian wildfire episode. This study indicated lower solar radiation on the ground (up to 80-150 $Wm^{-2}$ in diurnal averages), with a major reduction in the near-surface air temperature (between 0.2 and 2.6° on a regional scale over most of eastern Europe). Likewise, a reduction in the atmospheric boundary layer height (from 13% to





65%) and the vertical wind speed (from 5% to 80%) was found when ARI were included. Similar results have been reported by Baró et al. (2017) when the WRF-Chem model (Grell et al., 2005) was used to evaluate the effects of ARI and ACI.

Moreover, Forkel et al. (2016) analysed the different chemistry and physics options of WRF-Chem (Grell et al., 2005) and COSMO-MUSCAT (Wolke et al., 2012) (the same online coupled simulations used in this work) to evaluate the effects of ARI on radiation and temperature during both episodes studied herein. During the fires episode, the inclusion of this effect led a reduction between 10 and 100 $\mathrm{Wm}^{-2}$ in the average downward short-wave radiation at the ground level and a drop in the mean temperature of almost $1°$K over the area where the fires took place. During the dust outbreak episode, the ARI effect resulted in lower mean temperatures (-0.25°K over the Mediterranean Region). However, none of the aforementioned studies has evaluated the representation of aerosol optical properties and the effects of ARI+ACI on these properties.

Due to the uncertainty of aerosol effects, the above-mentioned coupling processes are treated differently in online coupled chemistry-meteorology models. This fact highlights the need to study the response of this type of models to the same aerosol emissions. Although the scope of this work is not to study ensemble forecasting, an ensemble mean was performed by using a set of simulations. The reason was because the ensemble mean is likely to provide the most skilful simulations compared to individual ones (Baklanov et al., 2014), like the results found by Fernández et al. (2009); Knutti et al. (2010); Kjellström et al. (2011). These results can be explained by the paradigm of different models being considered independent samples from a given distribution that is centred on the truth (Annan and Hargreaves, 2010). Hence the ensemble mean improves compared to the performance of individual models, and this ensemble could be expected to converge to the truth as more models are added to the ensemble.

Hence the main objectives can be summarised as: (1) assessing the representation of aerosol optical properties by a set of online coupled chemistry-climate simulations; (2)determining whether the inclusion of aerosol radiative feedbacks in this type of models improves the modelling outputs of aerosol optical properties over Europe. In order to achieve these objectives and to characterise the high uncertainty of the measured data, the first step was to determine the "best" satellite data according to AERONET to assess simulations. Afterwards, the representation of the aerosol optical properties by regional online coupled models was evaluated by comparing the ensemble of the models against this "best" satellite estimation.

## 2 Data and Methods

Aerosol optical properties considered for the evaluation in this work were aerosol optical depth (AOD) and Ansgtrön exponent (AE) at different wavelengths. These optical properties were assessed during two different episodes with a high aerosol load over Europe in 2010, as established in COST Action ES1004 EuMetChem. They were identified from the previous experience of phase 2 of the AQMEII modelling inter-comparison exercise (Galmarini et al., 2015), and consists in: the Russian wildfires and heat wave episode (from 25 July to 15 August 2010) and a Saharan desert dust outbreak with enhanced cloud and rain (from 2 to 15 October 2015).



## 2.1 Model simulations

The used simulations were run by working group 2 (WG2) of COST Action ES1004 EuMetChem (European framework for online integrated air quality and meteorology modelling, see http://www.eumetchem.info/). WG2 of EuMetChem investigated the importance of different processes and feedback in online coupled chemistry-meteorology models for air quality simulations
and weather forecasts.

The used simulations are summarised in Table 1. All these simulations, according to the model, chemical/physical options and feedbacks used, were denoted as an experiment. Two different online coupled models were used: COSMO-MUSCAT (Wolke et al., 2012) and WRF-Chem, version 3.4.1 (Grell et al., 2005) with different configurations (CS1, CS2, ES1 and ES3).

Simulations covered two different episodes with a high aerosol load over Europe in 2010. The Russian wildfires episode
lasted 22 days and the Sahara desert dust outbreak lasted 14 days. Both episodes were simulated following the common strategy for AQMEII phase 2, a sequence of 2-day time slices in which the meteorology is reinitialised every 2 days and the chemical state is adopted from the final step of the previous time slice (Galmarini et al., 2012, 2015, 2017). A spin-up of 5 days was used for the chemistry.

The target domains covered Europe. However for an easy clear analysis of aerosol effects, the assessment was done in a
window of the larger domain (see figure 1). The majority of the simulations were carried out with a grid width of approximately 23 km. Exceptions were CS2 (9.9 km) and DE3 ($0.125°$, around 14 km), both with higher resolutions for the Russian wildfires episode. The outputs of all the simulations were interpolated to a common lat-lon regular grid at a resolution of $0.1°$. The analysis grid for the Russian wildfires/heat wave case (figure 1, blue box) covered between $40°$ and $60°$, north and $20°$ and $60°$ east, with a grid size of 50,000 cells; that of the Saharan desert dust outbreak (figure 1, green box) covers $25°$ and $55°$, north
and $-10°$ and $30°$ east, with a grid size of 120,000 cells.

The simulations were run for three different feedback configurations: a base case called No Radiative Feedbacks (NRF), which does not take into account the radiative feedbacks of atmospheric aerosols in meteorology; a case that only bears in mind aerosol-radiation interactions (ARI); and another case that considers aerosol-radiation and aerosol-clouds interactions (ARI+ACI).

The meteorological initial and boundary conditions came from 3-hourly data with $0.25°$ of resolution from the European Centre for Medium-Range Weather Forecasts (ECMWF) operational archive. The chemistry boundary conditions for the main trace gases and particulate matter concentrations are available from the ECMWF IFS-MOZART model run in the MACC-II project (Inness et al., 2013, Monitoring Atmospheric Composition and Climate) with a grid width of $1.125°$ and a 3-hourly temporal resolution.

Anthropogenic emissions details can be found in Im et al. (2015a, b). They came from the TNO (Netherlands Organization for Applied Scientific Research) MACC emissions inventory (Pouliot et al., 2012; Kuenen et al., 2014; Pouliot et al., 2015, http://www.gmes-atmosphere.eu/) with a spatial resolution of $\bar{7}$ km. Annual emissions of methane ($CH_4$), carbon monoxide (CO), ammonia ($NH_3$), total non-methane volatile organic compounds (NMVOC), nitrogen oxides ($NO_x$), particulate matter ($PM_{10}$ & $PM_{2.5}$) and sulphur dioxide ($SO_2$) were made available by ten activity sectors. As part of the AQMEII and EuMetChem





initiatives, temporal profiles (diurnal, day-of-week, seasonal) were provided from Schaap et al. (2005) and vertical distributions were also made available.

The biomass burning emission data of the total PM emissions with a spatial resolution of $0.1°$ were estimated from the project IS4FIRES (Sofiev et al., 2009, Integrated monitoring and modelling system for wild-land fires). Other biomass burning
emission species were estimated as detailed in Im et al. (2015b). No heat release due to fires was taken into account.

In the COSMO-MUSCAT simulations, biogenic emissions were treated with the model described in Guenther (1993). Dust emission and transport were computed on the basis of meteorological and hydrological fields from COSMO (Heinold et al., 2007). In the WRF-Chem simulations, MEGAN (Guenther, 2006, Model of Emissions of Gases and Aerosols from Nature) was online coupled with WRF-Chem to estimate the biogenic emissions. Dust emissions were modelled according to Shaw
et al. (2008), with an adjustment made to avoid extremely high desert dust fluxes (Forkel et al., 2015).

For the COSMO-MUSCAT simulations, the physics options were: $\delta$-2-stream for long-wave and short-wave fluxes (Ritter and Geleyn, 1992); prognostic turbulent kinetic energy (TKE) planetary boundary layer (PBL); a multi-layer version of the former two-layer soil model after Jacobsen and Heise (1982); the Tiedtke (1989) mass-flux convection scheme. More details for the physical parametrisation are published in Doms et al. (2011). The cloud microphysics and chemistry options are
summarised in Table 1.

COSMO-MUSCAT takes into account ARI following Helmert et al. (2007). Radiative fluxes are computed online with the modified COSMO radiation scheme (Ritter and Geleyn, 1992) by considering variations in the modelled size-resolved aerosol fields. Radiation effects can influence the COSMO meteorology and feedback on emission and aerosol transport. No COSMO-MUSCAT simulations that took into account ACI were used in this work.

The common physics options for the WRF-Chem simulations were: the Rapid Radiative Transfer Method for Global (RRTMG) long-wave and short-wave radiation schemes (Iacono et al., 2008); the Yonsei University (YSU) PBL scheme (Hong et al., 2006); the NOAH land-surface model (Chen and Dudhia, 2001); the Grell 3D ensemble cumulus parameterisation (Grell and Dévényi, 2002). The cloud microphysics and chemistry options are summarised in Table 1. However, a more detailed description of the WRF-Chem simulations can be found in Forkel et al. (2015); Im et al. (2015a, b).

ARI are treated in WRF-Chem following Fast et al. (2006); Chapman et al. (2009); Barnard et al. (2010). Each chemical constituent of the aerosol was associated with a complex index of refraction. The overall refractive index for a given size bin was determined by volume averaging. The Mie theory and the summation overall size bins were used to determine the composite aerosol optical properties. Wet particle diameters were used in the calculations. Chapman et al. (2009) also described ACI. These interactions were implemented by linking the simulated cloud droplet number with the short-wave radiation and
microphysics schemes. Therefore, the droplet number affected both the calculated droplet mean radius and the cloud optical depth when using the short-wave radiation scheme. One limitation of WRF-Chem in the treatment of aerosol-cloud interactions was that these couplings were not computed in the convective clouds simulated by the cumulus parametrisation (Chapman et al., 2009; Yang et al., 2011; Schultz, 2016; Palacios-Peña et al., 2017).



## 2.2 Observational Data

The observational data used to evaluate the representation of the aerosol optical properties by the EuMetChem simulations came from different remote-sensing instruments: ground-based data from AERONET (Aerosol Robotic Network) and different sensors on board a satellite: the twin MODIS (Moderate Resolution Imaging), OMI (Ozone Monitoring Instrument) and
SeaWIFS (Sea-viewing Wide Field-of-view Sensor).

The data used from AERONET were AOD at 670nm and AE between 440/870nm from the available European stations during these episodes. Typically, the total uncertainty for the AOD data under cloud-free conditions is $< \pm 0.01$ for $\lambda > 440$ nm and $< \pm 0.02$ for shorter wavelengths (Holben et al., 1998).

For MODIS, the data used were the Level 2 of the Atmospheric Aerosol Product (MxD04_L2) from collection 6 (C6) with
a resolution of 10 km. Data came from the two available MODIS platforms, Aqua (MYD04_L2) and Terra (MOD04_L2). The MODIS data were estimated by two different algorithms, Dark Target (DT) and Deep Blue (DB). The DT algorithm variables used were: AOD at 550 nm for both ocean (estimated error (EE) $-0.02 - 10\%, +0.04 + 10\%$ [http://darktarget.gsfc.nasa.gov/products/ocean]) and land (EE $\pm 0.05 + 15\%$ [http://darktarget.gsfc.nasa.gov/products/land-10]); and AE between 550 and 840 nm over the ocean (preliminary EE is $0.45$ on pixels with an AOD $> 0.2$)(Levy et al., 2013). The DB algorithm used were:
AOD at 550 nm over land, with an EE approximately $\pm 0.05 + 20\%$ (Hsu et al., 2013; Sayer et al., 2013); and AE between 412 and 470 nm over land. Finally, a combined variable of the DT and DB algorithms with a wider coverage was used. This was AOD at 550 nm for both ocean and land, and its error has not yet been estimated (Levy et al., 2013).

The daily level 3, gridded at the $0.25°$, data from the algorithm called Multi-Wavelength of OMI sensor were also used. The used products were AOD at different UV range wavelengths (342.5 nm, 388 nm, 442 nm). The typical values for the retrieval
error of the AOD obtained by the non-linear fitting routine were below 0.03 (Team, 2009).

Finally, the daily level 3, gridded at $0.5°$, data from the SeaWIFS sensor were used. As with MODIS, this sensor employed the DB algorithm; consequently the error of these products was the same (Hsu et al., 2013; Sayer et al., 2013). The used products were AOD at different wavelengths over land (412 nm, 490 nm and 670 nm), over oceans (510 nm, 670 nm and 865 nm), and both (550 nm); and AE over land between 412/490 nm and over oceans between 510/670 nm.

## 2.3 Evaluation methodology

The first evaluation step consisted in establishing the "best" satellite data to assess simulations; for this purpose, different satellite data sets were compared with the AERONET observations by calculating the linear regression and the coefficient of correlation between the daily data. Afterwards, the evaluation of the model outputs against the selected satellite database was done by using classical statistics as the individual model-prediction error or bias ($e_i$), the mean bias error (MBE), the mean
absolute error (MAE), the coefficient of correlation ($r$) and the coefficient of determination ($r^2$) according to Willmott et al. (1985), Weil et al. (1992) and Willmott and Matsuura (2005).

Satellite data and model data were bilinearly interpolated to a common working grid, which corresponded with the analysis grid (described above) according to case studies. After the interpolation, and in order to compare with AERONET, the satellite





and simulations data were extracted from the cell that covered the corresponding station coordinates of the AERONET station by a nearest neighbour approach. The evaluation was done using both available the daily and hourly data.

Finally, to evaluate whether the inclusion of the radiative feedbacks in the simulations would lead to an improvement in the error of the model, two variables were defined: the *Normalized Improvement of the MAE* (Eq. 1) and the *Normalized*

5 *Improvement of the Coefficient of Determination* (Eq. 2), both in %. These variables were described so that a positive value would indicate an improvement due to the inclusion of the aerosol radiative feedbacks and a negative value; e.g., worsening. $|e_i|$ was the absolute error of the simulations.

$$\frac{1}{n}\left(\frac{\sum_1^n |e_i|_{NRF} - \sum_1^n |e_i|_{ARI/ARI+ACI}}{\sum_1^n |e_i|_{NRF}}\right) \times 100 \tag{1}$$

$$\frac{1}{n}\left(\frac{R^2_{ARI/ARI+ACI} - R^2_{NRF}}{R^2_{NRF}}\right) \times 100 \tag{2}$$

## 10  3   Results and discussion

The first part of this section identified the satellite data with the best skill according to the AERONET observations, which were considered referential. Then according to these results, the selected data were used to evaluate the representation of the aerosol optical properties by the different EuMetChem simulations during the Russian heat wave and wildfires episode, described in the second part of this section, and the Sahara desert dust outbreak, described in the third part.

## 15  3.1   Satellite-AERONET comparison

The results of the linear regression between the daily mean of the different satellite sensors and the AERONET data are shown in this subsection. Table 2 lists the correlation coefficient values for this statistical analysis. AE was excluded from this analysis as data were not available, which made the evaluation statistically non-significant.

In general, the best skills of the evaluation of the satellite products against AERONET were found for MODIS, as also

20 indicated in Myhre et al. (2005) or Bibi et al. (2015). Focusing on the results obtained during the Russian wildfires episode, the best skills were obtained for the DB algorithm of MODIS (correlation values over 0.80 both from the Terra and Aqua platforms). Indeed the best AOD estimation was obtained from the DB algorithm of MODIS from the Terra platform, with a $r^2$ of 0.82. The aerosol products retrieved by the DB algorithm provide useful information about aerosol properties over bright-reflecting land surfaces, such as desert, semiarid and urban regions. In C6, the algorithm was enhanced in order to improve

25 retrievals over regions of mixed vegetated and non-vegetated surfaces (Hsu et al., 2013), and to cover all the cloud-free and snow-free land surfaces (Sayer et al., 2014). This fact could explain the good representation of this satellite product over the study area. The best estimated variable from the OMI data was AOD at 342.5 nm with a correlation value of 0.77. For this episode, no data from the SeaWIFS sensor were available.



During the Saharan dust episode, the best skills of the AOD satellite products were found for the combined and DT algorithm products of MODIS from the Aqua platform. The correlation values were much higher than for the other satellite products ($r^2 = 0.92$). The DT algorithm was composed of two different algorithms, which were used to retrieve aerosol information over land (dark at visible and longer wavelengths) and over vegetated/dark-soiled land (dark at visible ones) (Sayer et al.,

2014). This provided data over most of Europe. Its combination with the enhanced DB algorithm in C6 provided expanded coverage, which covered most of the domain. The OMI products estimation presented lower correlation values (0.26) than for the Russian wildfires episode.

SeaWIFS sensor data were available during the Saharan dust episode. The evaluation of the AOD estimation at $670nm$ showed that this sensor produced better estimations over oceans (correlation values of 0.90) than over land (0.53). SeaWIFS

data were partnered with an expanded DB algorithm and the SOAR (SeaWiFS Ocean Aerosol Retrieval) algorithm, with retrievals over oceans and inland water bodies (http://disc.sci.gsfc.nasa.gov/gesNews/swdb_monthly_in_Giovanni). Thus using this combination improved retrievals over oceans, as our results showed.

Therefore, in order to evaluate the representation of aerosol optical properties by the modelling experiments for the Russian wildfires episode, the variables of the DB algorithm from MODIS aboard the Terra Platform were used. For the Saharan dust

episode, the combined (DT and DB algorithms) AOD at $550nm$ and the AE of the DT algorithm from MODIS aboard the Aqua platform were selected.

### 3.2   Evaluation of simulations

Once the most skilful satellite data were defined, these data (together with AERONET observations) were used to establish the biases and errors of the modelling experiments for the episodes covered in this work.

### 3.2.1   Russian wildfires case

The evaluation of the experiments for the Russian wildfires and heat wave episode are detailed in this subsection. The variables of the DB algorithm from MODIS aboard the Terra Platform were used. In order to show more confident results, a mask that showed the areas where the satellite observations were higher than the 30% of their maximum, was implemented. Figures 2 shows the evaluation of AOD at $550nm$ and Figures 4 of AE between $412nm$ and $470nm$. From the AERONET data, Figure

3 displays the results for AOD at $670nm$ and Figure 5 for AE between $440/870nm$.

Figure 2a shows the values of AOD measurements by MODIS. The highest values around 2.6 were found over Russia and its surroundings areas due to emissions from the wildfires. The estimation of the mean bias error (MBE) is shown in Figure 2b. All the WRF-Chem simulations (CS1, CS2, ES1, ES3) and the ensemble mean underestimated AOD over the fire-affected areas (minimum MBE values for NRF in Table 3). Over the rest of the domain, a low overestimation (around 0.4, see the maximum

MBE values in Table 3) was obtained with the WRF-Chem simulations and the ensemble mean. For DE3, the underestimation was lower and did not cover such a large area as the rest of the experiments. However, a higher overestimation was found in DE3 over the rest of the domain.





WRF-Chem tended to underestimate high AOD values, so these AOD levels were lower than the MODIS levels, especially over the areas with high aerosol loads. When comparing the WRF-Chem experiments, following a modal aerosol approach (CS1, CS2, ES1) led to lower error values than a sectional approach (ES3). Conversely, COSMO-MUSCAT tended to present higher AOD values than WRF-Chem. Thus its underestimation was lower than WRF-Chem, but sometimes led to a high

overestimation of the satellite-observed AOD. As mentioned in the Introduction, the ensemble mean generally presented the best skills, as stated for other variables in different works (e.g. Baklanov et al., 2014; Fernández et al., 2009; Knutti et al., 2010; Kjellström et al., 2011).

Generally for the ARI and ARI+ACI simulations, slightly lower MBE values than NRF were found in all the experiments (e.g., in the ES1 simulations: NRF −1.7; ARI −1.66; ARI+ACI 1.64). However, the MBE for the ensemble (NRF −1.55; ARI

−1.49; ARI+ACI −1.71) did not show this improvement. Its analysis should be carefully taken into account because the ARI ensemble mean did not include the CS1 simulation and the ARI+ACI, the DE3 simulation.

The MBE indicates the bias of the model error (tendency to over- or underestimate satellite observations). The mean average error (MAE, Table 3) indicates the magnitude of the model errors. The maximum MAE values over Russia and the surrounding areas were similar or came close to the maximum or minimum MBE values, which indicated that the biases observed above

moved in the same way (positive or negative) throughout the simulations period.

The estimated *Normalized improvement of MAE* (Figure 2c) indicates whether the inclusion of aerosol radiative feedbacks reduces the magnitude of model errors; i.e., , whether the inclusion of ARI or ARI+ACI improves the skills of the models as regards the absolute error. The results indicated a generalized improvement of the MAE over the whole domain. For ARI, the ensemble mean presented an improvement above 30%, but the simulations with the greatest normalized improvement were ES1

(> 40%, over a large area of the domain) and CS2 (normalized improvement values up to 90%). Moreover, these improvements took place especially over the areas with high AOD values (areas with fires). For ARI+ACI over the same area, the ensemble mean presented an improvement of up to 45%. However in the southwest area of the domain, improvements reached 70%. Over the fire-affected areas, the experiment that led to major improvement was CS1 (> 50%).

Regarding the determination coefficient ($r^2$, Figures in Appendix, which only show the areas where correlation results were

significant at 90%), no clear spatial pattern was associated with this parameter or with its improvement. However, over the whole domain the ensemble mean presented higher values than each specific experiment for $r^2$, and a clear improvement in the determination coefficient was observed when radiative feedbacks were taken into account (ARI+ACI).

The hourly AOD values recorded at 670nm of the different simulations correlated with the AERONET values at this same wavelength. Only the correlation results that were significant at the 95% level in a certain station are shown in Figure 3a. The

Model-AERONET $r$ values were > 0.4 in nearly 75% of the stations, and were above 0.5 in nearly 50% of the stations used for all the experiments.

The station where the NRF, ARI and ARI+ACI ensemble means showed the best skills was the Toravere station (correlation coefficient for NRF of 0.68, ARI 0.73 and ARI+ACI 0.73, Figure 3b). At this station (located in northerly areas) AOD was higher between 25 and 30 July, and between 5 and 10 of August. However, the maximum correlation values among all the

experiments were found at the Efoire station (Figure 3c) for the CS2 configuration (correlation coefficient for NRF 0.83, ARI





0.84 and ARI+ACI 0.82). This can be explained by the enhanced resolution of CS2 around 9.9km. This fine resolution may lead to improvements in the local representation of AOD. It should be highlighted that no clear improvement in the model-AERONET correlations was noted when the aerosol radiative feedbacks were taken into account.

As indicated Palacios-Peña et al. (2017), the AOD discrepancies between simulations and observations can be attributed to
the errors in the model estimations of the aerosol dry mass, the fraction of particles for a given mass or water associated with aerosols. According to the aerosol dry mass estimation, Im et al. (2015b) found an overestimation in the simulated $PM_{2.5}$ levels for the AQMEII phase 2 simulations (comparable to EuMetChem simulations) and Soares et al. (2015) established a rough overestimation at about 50% of the total biomass the burning emissions used here. According to this evidence, an overestimation of the AOD levels over the fires area can be expected for this episode. However, our results showed an AOD
underestimation. This behaviour could be due to the understated injection height of the total biomass burning emissions found by Soares et al. (2015). In other words, a misunderstanding of the simulation of the aerosol vertical distribution could affect the representation of the aerosol optical depth. We established this statement following Kipling et al. (2016), which found the extent to which the biomass burning emissions to the free troposphere affected the vertical profile of aerosols, causing changes in AOD and radiative forcing with the HadGEM3-UKCA model.

The MODIS AE satellite values are shown in Figure 4a. AE is a variable that provides an idea of particle size in the atmosphere. High AE values indicate fine particles and low values denote coarse particles. High values between 1.7 and 1.8 were found over Russia and the surroundings areas given the fine particles emitted by wildfires. Low AE values were found over the southeast (SE) areas of the domain. This fact could be explained by the presence of coarse particles from the mineral dust that came from nearby desert areas.

A general MBE overview (Figure 4b)indicated that all the experiments underestimated high AE values and overestimated low AE ones. Hence the model underestimated the variability of this variable. Over the Russian wildfires area, where the highest AE values were found, simulations underestimated (minimum MBE values for NRF in Table 3) this variable. Over the area with the lowest AE values (the SE area of the domain), the model produced an overestimation (maximum MBE values in Table 3). According to these results, and in the same way as AOD, the most skilful simulation was the ensemble mean. ES3
(WRF-Chem simulations using a sectional approach) presented higher AE values (lowest minimum and highest maximum MBE) than the other experiments. No simulations for the DE3 experiments were available for AE.

Figure 4(c) shows the MAE values and the improvements for AE. Maximum MAE values (Table 3) were found over the south-east area of the domain. Only ES3 experiment, which used the aerosol sectional approach, presented an improvement to AE representation when ARI+ACI were taken into account (Figure 4c, right column). This improvement, with values around
20%, was found over the areas where the MAE values were lower.

The determination coefficient values (Appendix), showed only the areas where correlation results were significant at 90%, of the evaluation made of AE were worse than they were for AOD. The highest values (>0.4) were observed over the Russian wildfires areas. Values below 0.25 were found for the rest of the domain. Although the minimum MAE values were shown for ES3, this experiment (which considered aerosols with a sectional approach) presented a worse determination coefficient over





the Russian wildfires area than the other experiments. Despite there being no clear improvement pattern for the determination coefficient, the ARI+ACI scenario showed isolated improvements with values between 50% and 100%.

Regarding the model-AERONET comparison (Figure 5a),the number of stations with available AE data was very limited and substantially lower than for AOD. Likewise, the correlation coefficient values were lower for AE. Between 10% and 40%

of the used stations obtained model-AERONET correlations above 0.5, depending on the model used. The largest number (> 40%) of stations with correlation coefficients >0.5 was found for the ensemble mean. The station where the ensemble mean had the most skills was Bucharest (correlation coefficient for NRF 0.72, ARI 0.69 and ARI+ACI 0.74, Figure 5b). For this variable, no station had higher correlation coefficient values for any specific experiment. As for AOD, the inclusion of aerosol radiative feedbacks did not improve the AE representation.

### 3.2.2 Saharan desert dust outbreak case

This section describes the evaluation of the representation of AOD and AE by the simulations done for the Saharan dust episode (Figures 6 to 9). In this case, the combined (DT and DB algorithms) AOD at $550nm$ (Figure 6) and the AE of the DT algorithm (Figure 8) from MODIS aboard the Aqua platform were used. As with the Russian wildfires episode, the mask of 30% of the maximum of observations was implemented. The model-AERONET comparisons of AOD at $670nm$ (Figure 7) and AE

between $440/870nm$ (Figure 9) were also shown.

Figure 6 shows the AOD values measured by MODIS. Over the southeast are of the domain, values of >0.6 were observed due to transported dust. This value was not very high for a dust outbreak, but was caused by wet deposition (rain during the episode) and was explained in the event description in the section "Model simulations".

In Figure 6b, all the experiments (CS2 simulations were not available in this case) underestimated high AOD values (over

the southeast area of the domain). The MBE values over this area were around -0.3 for DE3, but were lower (around -0.2) for the other experiments (WRF-Chem simulations). However small areas with a higher underestimation were found over this zone (minimum MBE values for NRF in Table 3). Over the rest of the domain, minor overestimations were modelled (MBE values around 0.1). Conversely, small sporadic areas with high overestimations were found (maximum MBE values for NRF in Table 3).

In order to consider the magnitude of these under/overestimations, the MAE is depicted in Figure 6c (left column). Over the southeast area of the domain, the MAE gave values around 0.2 for all the experiments, except for DE3, which showed an MAE value >0.3. The most relevant improvements to this statistic (Figure6c, centre and right column) were observed for the ARI+ACI ensemble mean, with improvements up to 25%, over desert areas. Smaller areas over the Mediterranean Sea with improvements up to 75% were found for the ES3 simulation. These improvements could be explained by a better representation

of the coarse mode in the sectional approach.

The determination coefficient (Appendix) values were higher than for the Russian wildfires episode. These values came close to 1 for all the experiments over most of the domain. No improvement in the representation of $R^2$ when ARI and ACI were taken into account was observed. For both variables AOD and AE, the established requirement of showing only the areas where correlation results were significant at 90% was implemented.



Both WRF-Chem and COSMO-MUSCAT tended to underestimate high AOD values. COSMO-MUSCAT was the model with the highest underestimation. Regarding the WRF-Chem simulations, ES3, which used the sectional approach, presented higher AOD values (lower minimum and higher maximum MBE values) than the other experiments. According to Im et al. (2015b), this underestimation can be explained by a systematic underestimation for all the models at the $PM_{10}$ levels with

the highest underestimation for the Mediterranean Region. Moreover, this major underestimation can be attributed to natural emissions, such as the desert dust studied in this episode.

Regarding the model-AERONET comparison (Figure 7a), the ensemble mean showed the best skills at the Toulon station (correlation values 0.98, 0.98 and 0.96 for NRF, ARI and ARI+ACI, respectively, Figure 7b). According to these values, it was not possible to establish an improvement when aerosol radiative feedbacks were taken into account. Besides, more than 60%

of the stations used gave correlation values above 0.5 for all the experiments. Around 80% of the used stations presented this behaviour for ES1. For these experiments, the highest correlation values were above 0.90 at stations like Toulon (Figure 7b), Efoire (Figure 7c) or Thessaloniki (not shown).

Although the number of data per station was smaller for this episode than for the Russian wildfires episode, all the shown stations complied with the established requirement of the correlation results, which were significant at the 95% level (explained

in the previous section).

For AE, the MODIS satellite values are shown in Figure 8a. Values between 0 and 1 were found over the southeast area of the domain due to coarse dust particles. When the bias was analysed (Figure 8b, Table 3), ES3 (the aerosol sectional approach) displayed a different behaviour from the rest of the experiments. ES3 underestimated low AE levels over the area affected by the Saharan dust outbreak, with a minimum MBE for NRF of $-1.43$. An improvement to the AE representation

was also achieved for ARI+ACI, with a minimum MBE error of $-1.31$ (this could also be observed for the improvement to MAE, explained below). Small overestimations of the values with a maximum MBE of 0.33 were found for NRF. The other experiments exhibited similar behaviour as they did for the Russian wildfires episode. Low values (over the southeast area of the domain) were overestimated and high values were underestimated.

MAE (Figure 8c, left column) estimates the magnitude of errors. Maximum MAE errors (Table 3) were found over the dust

outbreak-affected area. Regarding improvements to this statistical figure shown in Figure 8c (centre and right column), the ensemble mean presented improvements of up to 50% over most of the domain when ARI were included. However, when considering ARI+ACI, ES3 had the highest improvement values with values of up to 75% over the dust outbreak-affected area.

As for AOD, the determination coefficient values (Appendix) were higher than for the Russian wildfires episode and came close to 1 over most of the domain. All the experiments provided these $R^2$ values, with no significant improvement to this

statistical figure when ARI and ACI were taken into account.

Finally, the ensemble and CS1 presented the best skills when they were compared with AERONET (Figure 9a). Athens (Figure 9b) was the station where the ensemble mean presented its best skill (correlation coefficients of 0.96, 0.91 and 0.92 for NRF, ARI and ARI+ACI respectively) and Helgoland (Figure 9c), done by CS1 simulations (correlation values for NRF 0.96 and ARI+ACI 0.95). The latter station did not present any strong dust outbreak influence. It was not possible to establish

a significant improvement when aerosol radiative feedbacks were taken into account. Notwithstanding, more than 40% of the



used stations presented correlation values above 0.5 for all the experiments. For the ensemble mean, correlation values above 0.5 were found from 70 to 80% of the used stations.

## 4 Summary and Conclusions

This work attempts to identify the skill of a set of satellite observations and online coupled models to represent the aerosol
optical properties during two important episodes: a biomass burning episode and a dust outbreak episode that occurred over Europe in 2010. The first task consisted in identifying the skills of different satellite observations compared with AERONET. In this sense, MODIS showed the highest correlation coefficient with the ground-based observations. The DB algorithm of MODIS provided the best skilful products for the Russian wildfires/heat wave case. This was because the algorithm was enhanced over regions of mixed vegetated and non-vegetated surfaces, like the study area. The SOAR algorithm also provided
a high quality product over oceans when it was used for the SeaWIFS sensor during the dust outbreak episode because SOAR was developed to obtain retrievals over oceans and inland water bodies. However, the most skilful AOD product for the dust outbreak episode was that provided by MODIS combined DB and DT algorithm. This combined product provided an expanded coverage over bright-reflecting land surfaces, such as desert, semiarid and urban regions, which was the case of this dust outbreak. For AE, there were not enough data to estimate how good its representation was.

Using the best-available satellite information, a set of regional online chemistry-climate/meteorology models over Europe were evaluated for the two aforementioned episodes. The results for the Russian wildfires episode indicated that AOD was better represented by the models than AE. AOD was generally underestimated by all the simulations. This behaviour could be explained by a misunderstanding in the aerosol vertical distribution simulation due to the understated injection height of the total biomass burning emissions, which could affect the representation of AOD. The underestimation by WRF-Chem
was higher than by COSMO-MUSCAT, and the latter simulated higher AOD values than all the WRF-Chem configurations. Among the WRF-Chem experiments, the modal aerosol approach (CS1, CS2 and ES1) presented lower error values than the sectional approach (ES3). The inclusion of all the aerosol radiative feedbacks (ARI+ACI) led to an improvement above 30% to the MAE of the ensemble mean. The ensemble mean was more skilful than the single simulations compared with the satellite products. However at a local level, the fire episode simulation at a fine resolution (CS2) presented the highest model-
AERONET correlations (around 0.8). No clear improvement to the model-AERONET correlations was found when ARI and ACI were included.

In the Russian wildfires case, the MBE, MAE and $R^2$ values were lower for AE than for AOD. All the experiments underestimated the variability of AE (i.e., low values were represented by too high simulated values and high values by lower values than actually observed). According to all the calculated statistics, the most skilful experiment was the ensemble mean. When
considering individual models, the MBE indicated that ES3 (the only WRF-Chem simulation that used a sectional approach) presented higher AE values than the other experiments, and thus resulted in a lower underestimation. Despite these higher values, ES3 was the only experiment to present improvements (around 20%) when ARI+ACI were taken into account. For



$R^2$ no clear improvement was noted when aerosol radiative feedbacks were included. For AE, the highest model-AERONET correlations were found for the ensemble mean and they were lower (around 0.7) than for AOD.

The Saharan desert dust outbreak case results indicated that high AOD values, which were found over the southeast area of the domain (dust-affected area), were underestimated by all the models and configurations. According to the results from Im et al. (2015b), this underestimation can be explained by an underestimation in the $PM_{10}$ levels, attributed mainly to natural emissions, such as desert dust. DE3 presented the most pronounced underestimation. The ES3 experiments, which used the sectional approach, estimated higher AOD values than the other simulations. The most obvious improvement to the skill from including ARI+ACI went to the ensemble mean, with an improvement up to 25%. However, we can find a better improvement over the Mediterranean Sea with values around 75%, which can be explained by a better representation of the coarse mode in the sectional approach. $R^2$ for AOD during the dust outbreak episode obtained values close to 1 over most of the domain. Although this statistical figure showed better results for this episode than for the Russian wildfires episode, no clear improvement for the aerosol radiative feedbacks was obtained. When models and AERONET were compared, the ensemble mean provided the best skill with correlation values above 0.9. More than 60% of the used stations presented correlation values above 0.5 for all the experiments. ES1 showed skills for correlations over 0.5 in 80% of the used stations, sometimes with correlations of >0.9.

Finally, the representation of AE in the dust outbreak case presented similar performance to that for the Russian wildfires episode. The modelling experiments underestimated the variability of this variable, with ES3 providing lower AE values than the other experiments. However for ARI+ACI, an improvement by up to 75% was found for the ES3 experiment. The ensemble mean offered the best skill and showed an improvement by up to 50% when ARI were taken into account. The AE determination coefficient was better for this episode than for the Russian wildfires episode. Locally, the model-AERONET comparison indicated that the ensemble mean and CS1 presented the best skills, with correlation values above 0.9. For the ensemble mean, correlation values above 0.5 were found for 70 to 80% of the used stations; this number was lower (around 40%)for the single experiments.

By way of conclusion, the modelling experiments generally underestimated high AOD values. According to Palacios-Peña et al. (2017), AOD discrepancies between simulations and observations can be attributed to errors in the model estimation of the aerosol dry mass, the fraction of particles for a given mass or the water associated with aerosols. A model underestimation of the aerosol dry mass of dust can explain the AOD underestimation during the Saharan desert dust episode because of the underestimation in the $PM_{10}$ levels found by Im et al. (2015b). However, this fact cannot explain the AOD underestimation obtained during the Russian wildfires episode, when a misunderstanding in the aerosol vertical distribution simulation, due to an understated injection height of the total biomass burning emission, described by Soares et al. (2015), could affect the AOD representation. Locally, fine resolutions can help improve the representation of this variable. For AE, the aerosol representation by the modal WRF-Chem approach resulted in a substantial underestimation in this variable's variability. The sectional WRF-Chem approach provided high AE values for the case with fine fire particles and low values for the case with coarse particles (dust). Thus the bin representation for different particle sizes strongly influenced the estimation of this variable when a sectional approach was used. Hence the use of a sectional approach improved the size distribution representation in the models, but led to a worse skill to represent AOD.





Albeit sometimes an individual experiment may show better results, as we assumed in the first section, the ensemble mean generally provides the most skilful simulation. Regarding the inclusion of aerosol radiative feedbacks, the observed improvements (especially for a sectional approach) justify not only the inclusion of radiative feedbacks in online coupled models, but also the increase in the computational time to include these effects. Further studies are necessary to improve the model representation of aerosol optical properties.

*Acknowledgements.* The authors acknowledge Project REPAIR-CGL2014-59677-R of Spanish Ministry of the Economy and Competitiveness and the FEDER European program for support to conduct this research. Work was possible thanks to the fellowship 19677/EE/14 funded by Fundación Séneca-Agencia de Ciencia y Tecnología de la Región de Murcia, Programme Jiménez de la Espada de Movilidad, Cooperación e Internacionalización, within the framework of II PCTIRM 2011-2014 and the fellowship FPU14/05505 funded by the Spanish Ministry of Educacition, Culture and Sport. The authors thank the support from EuMetChem COST ACTION ES1004 and the Air Quality Modelling Evaluation International Initiative (AQMEII). We also thank the researchers and their staff for establishing and maintaining the European AERONET sites used in this research in the network AERONET, and many others who have been involved in the MODIS, OMI and SeaWIFS datasets (NASA).



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



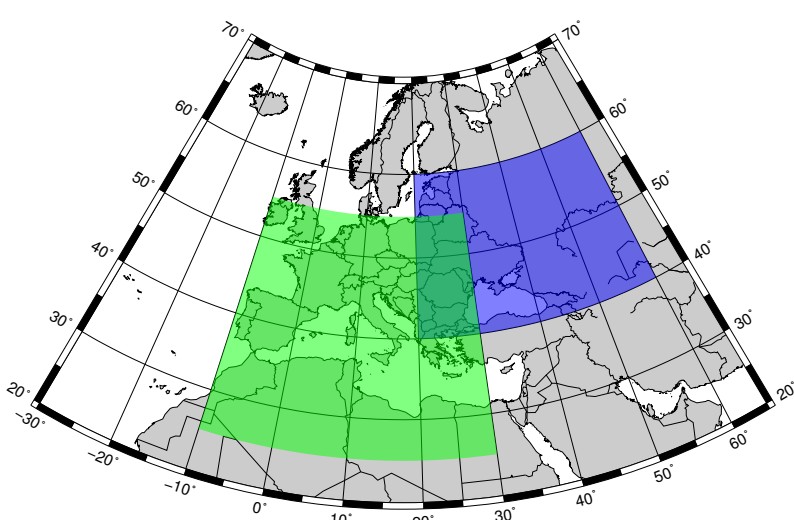

**Figure 1.** Target (grey) and analysis domains (blue for the Russian wildfires/heat wave case, green for the Saharan desert dust outbreak).





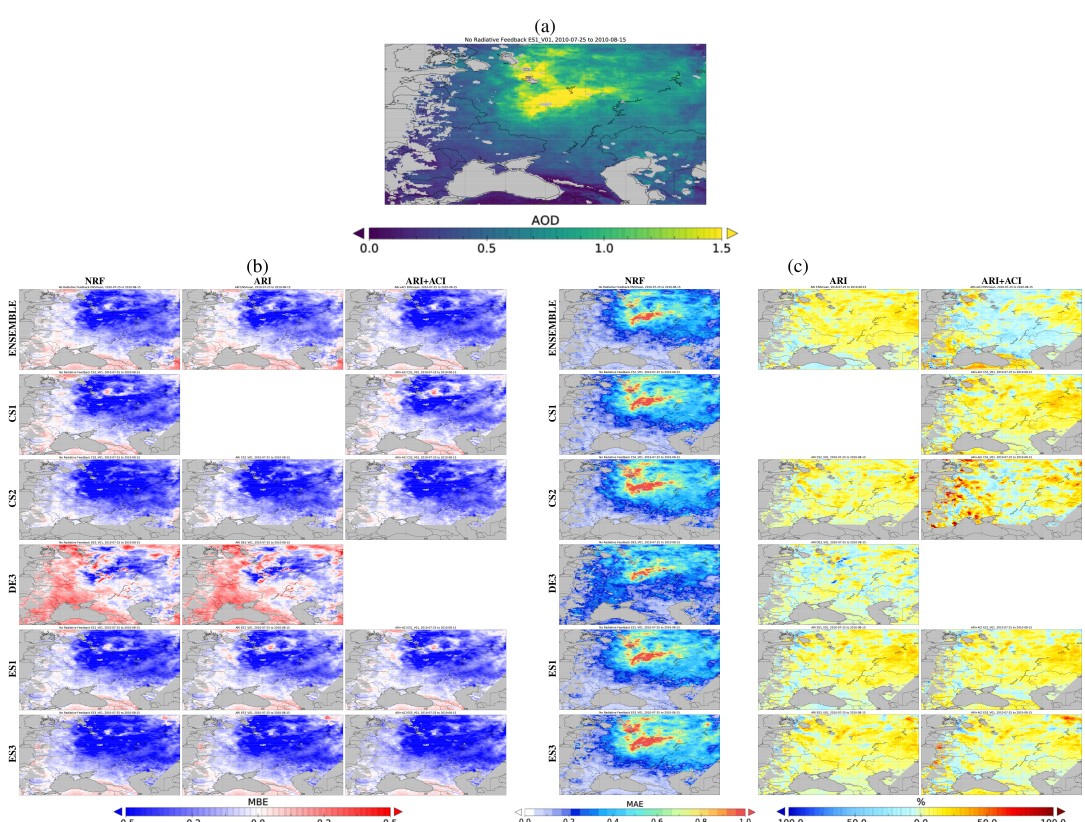

**Figure 2.** Model-MODIS comparison of AOD at $550nm$ for the Russian wildfires case: (a) Satellite values; (b) MBE for NRF (first column), ARI (second) and ARI+ACI (third); and (c) MAE for the NRF simulations (first column) and their improvements due to ARI (second) and ARI+ACI (third).





**Figure 3.** Model-AERONET comparison of AOD at $670nm$ for the Russian wildfires case: (a) temporal correlation values at each AERONET station; and temporal series at the (b) Toravere and (c) Eforie stations.





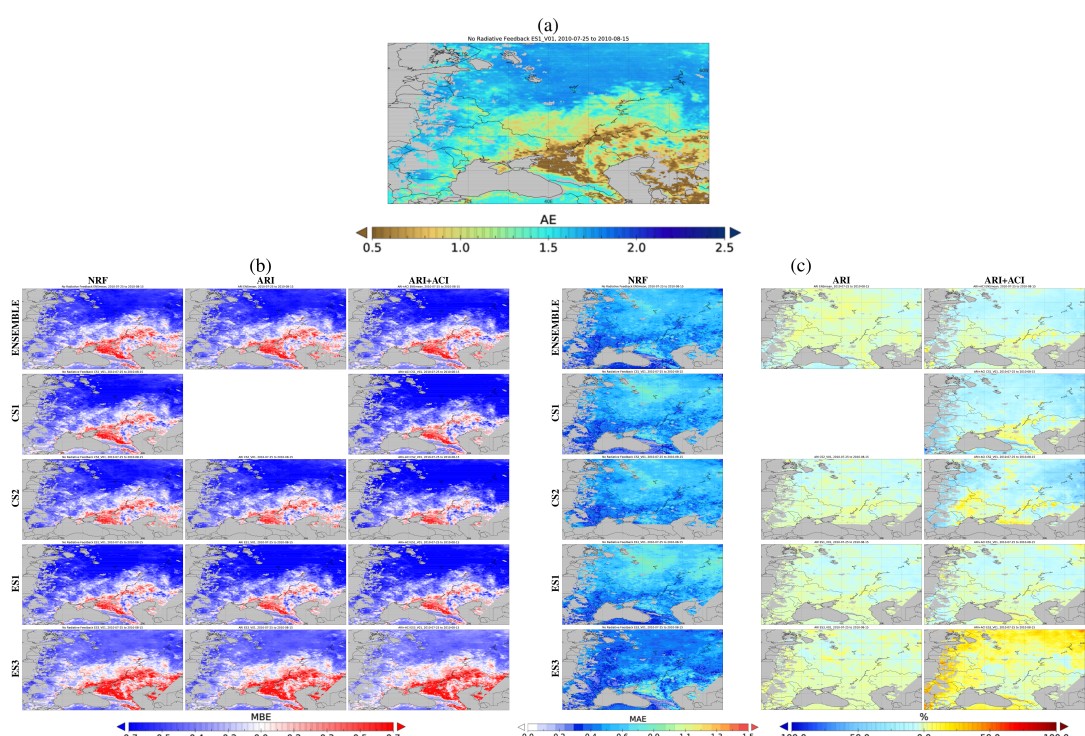

**Figure 4.** Model-MODIS comparison of AE between $412/470nm$ for the Russian wildfires case: (a) Satellite values; (b) MBE for NRF (first column), ARI (second) and ARI+ACI (third); and (c) MAE for the NRF simulations (first column) and their improvements due to ARI (second) and ARI+ACI (third).





**Figure 5.** Model-AERONET comparison of AE $440/870nm$ for the Russian wildfires case: (a) temporal correlation values at each AERONET station; and temporal series at (b) the Bucharest station.





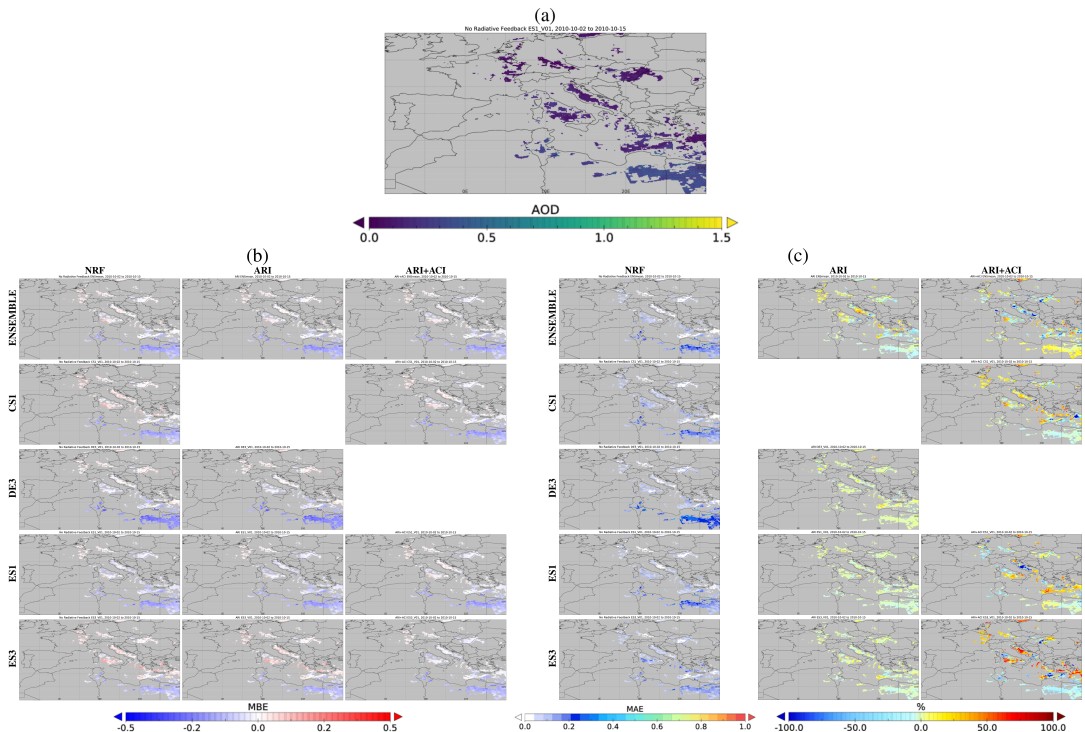

**Figure 6.** Model-MODIS comparison of AOD at $550nm$ for the Saharan desert dust outbreak case: (a) Satellite values; (b) MBE for NRF (first column), ARI (second) and ARI+ACI (third); and (c) MAE for the NRF simulations (first column) and their improvements due to ARI (second) and ARI+ACI (third).

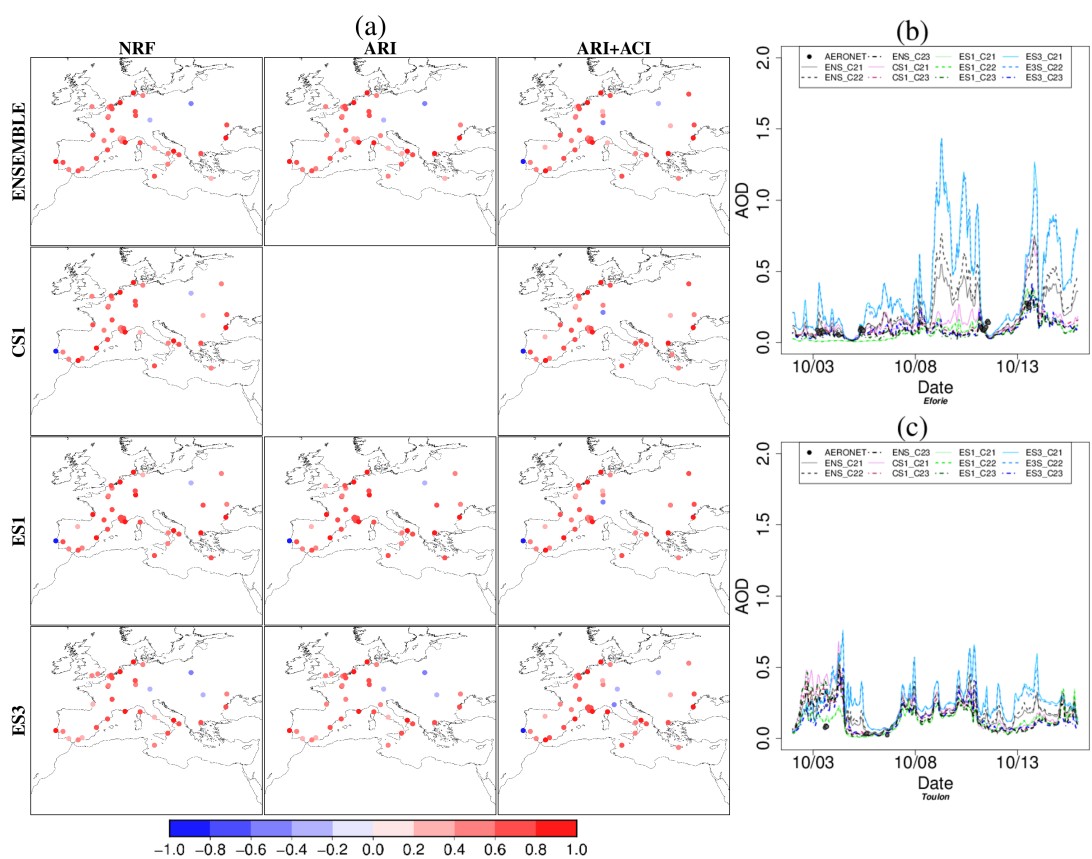

**Figure 7.** Model-AERONET comparison of AOD at $670nm$ for the Saharan desert dust outbreak case: (a) temporal correlation values at each AERONET station; and temporal series at the (b) Toulon and (c) Eforie stations.





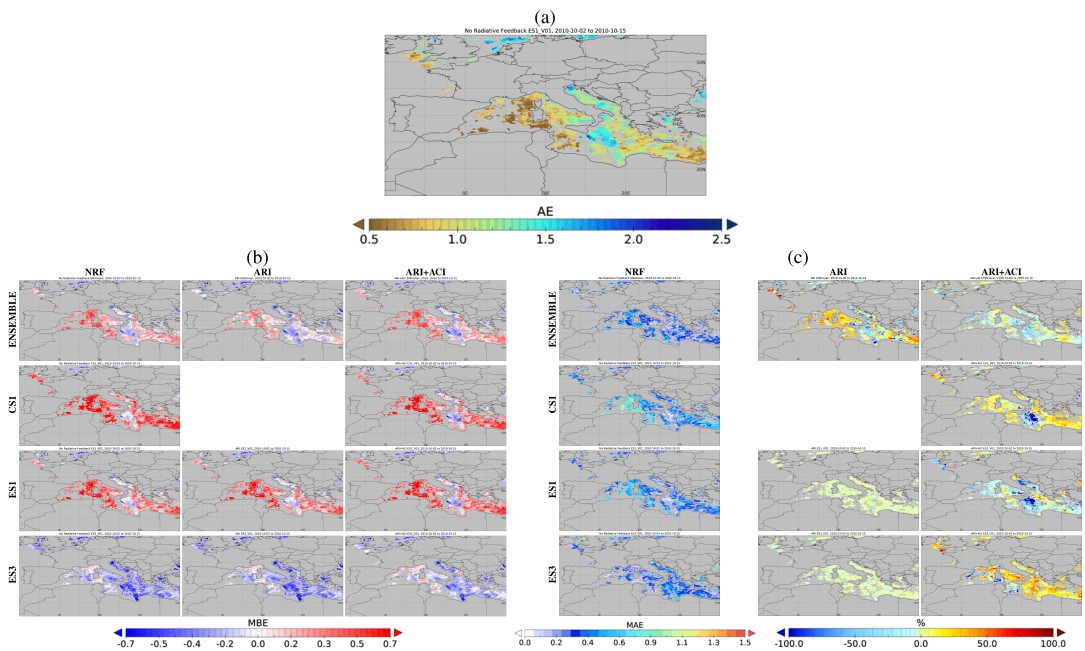

**Figure 8.** Model-MODIS comparison of AE between $550/860nm$ for the Saharan desert dust outbreak case: (a) Satellite values; (b) MBE for NRF (first column), ARI (second) and ARI+ACI (third); and (c) MAE for the NRF simulations (first column) and their improvements due to ARI (second) and ARI+ACI (third).



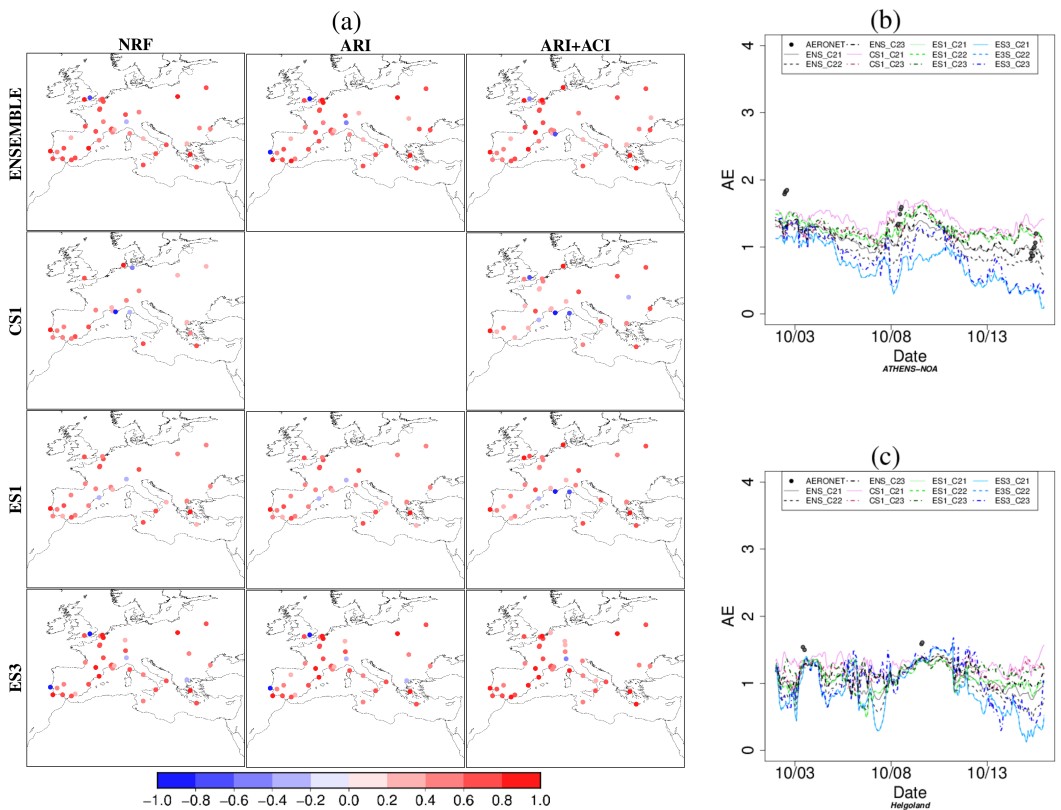

**Figure 9.** Model-AERONET comparison of AE $440/870nm$ for the Saharan desert dust outbreak case: (a) temporal correlation values at each AERONET station; and temporal series at the (b) Athens and (c) Helgoland stations.





**Table 1.** Model simulations

|  | CS1 | CS2 | DE3 | ES1 | ES3 |
|---|---|---|---|---|---|
| Lead Institution | KIT/IMK-IFU* | KIT/IMK-IFU* | TROPOS Leipzig | UMU-MAR | UPM-ESMG |
| Case | Russian wildfires & Dust event | Russian wildfires | Russian wildfires & Dust event | Russian wildfires & Dust event | Russian wildfires & Dust event |
| Runs** | NRF, ARI+ACI | NRF, ARI, ARI+ACI | NRF, ARI | NRF, ARI, ARI+ACI | NRF, ARI, ARI+ACI |
| Spatial Resolution | 23km | 9.9km | 0.125° (Russian wildfires); 0.25° (Dust event) | 23km | 23km |
| Model | WRF-Chem v.3.4.1 | WRF-Chem v.3.4.1. | COSMO-MUSCAT | WRF-Chem v.3.4.1. | WRF-Chem v.3.4.1. |
| Chemical option | RADM2 modified (Stockwell et al., 1990) | RADM2 modified (Stockwell et al., 1990) | RACM-MIM2 (Karl et al., 2006; Tilgner et al., 2008) | RADM2 (Stockwell et al., 1990) | CBMZ (Zaveri and Peters, 1999) |
| Aerosol option | MADE-SORGAM (Ackermann et al., 1998; Schell et al., 2001) | MADE-SORGAM (Ackermann et al., 1998; Schell et al., 2001) | Simpson et al. (2003) | MADE-SORGAM (Ackermann et al., 1998; Schell et al., 2001) | MOSAIC (Zaveri et al., 2008) |
| Microphysic option | Morrison et al. (2009) | Lin et al. (1983) | Kessler-type bulk (Doms et al., 2011) | Lin et al. (1983) | Morrison et al. (2009) |
| Wet deposition | Simple | Easter et al. (2004) |  | Easter et al. (2004) | Simple |
| Aqueous chemistry | - | Fahey and Pandis (2001) |  | Fahey and Pandis (2001) | - |

*joint effort with ZAMG, RSE and UPM-ESMG

**NRF: No Radiative Feedbacks, ARI: Aerosol-Radiation Interactions, ARI+ACI: Aerosol-Radiation and Aerosol-Clouds Interactions



**Table 2.** Coefficient of correlation of the comparison between satellites and the AERONET AOD data

| Case | MOD04_L2 | | MYD04_L2 | | OMI | | SeaWIFS | | | |
|------|----------|------|----------|------|------|------|------|------|------|------|
| Russian wildfires | Deep Blue | 0.82 | Deep Blue | 0.80 | $342.5nm$ | 0.80 | | | | |
| | Combined | 0.66 | Combined | 0.70 | $388nm$ | 0.76 | | | | |
| | Dark Target | 0.76 | Dark Target | 0.79 | $442nm$ | 0.75 | | | | |
| Dust outbreak | Deep Blue | 0.45 | Deep Blue | 0.64 | $342.5nm$ | 0.26 | $412nm\_L$ | 0.69 | $550nm$ | 0.74 |
| | Combined | 0.77 | Combined | 0.92 | $388nm$ | 0.26 | $490nm\_L$ | 0.69 | $670nm\_O$ | 0.90 |
| | | | | | | | | | $670nm\_L$ | 0.53 |
| | Dark Target | 0.77 | Dark Target | 0.92 | $442nm$ | 0.26 | $510nm\_O$ | 0.89 | $865nm\_O$ | 0.90 |

$L$: satellite product over land, $O$: satellite product over ocean

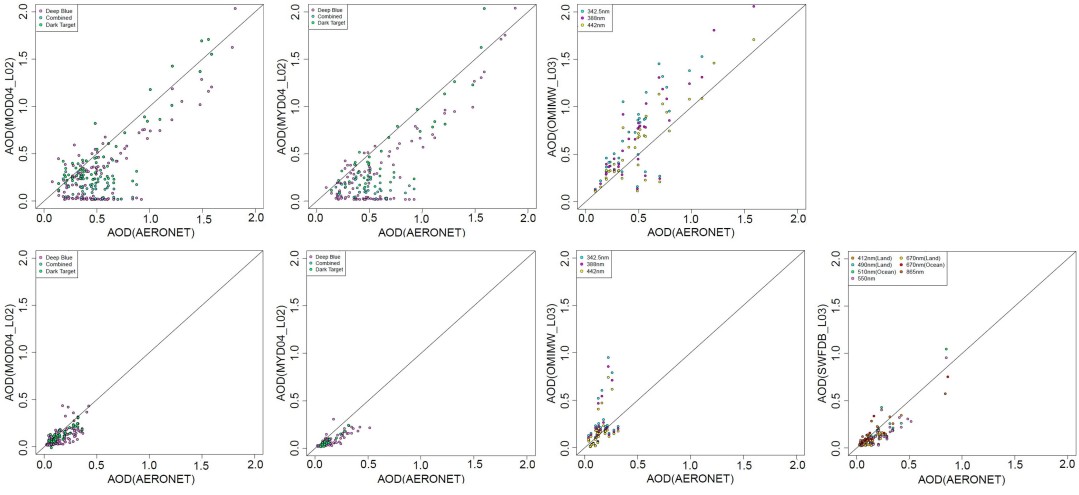

**Figure A1.** Satellite-AERONET linear regression. Russian wildfires case (top) and Saharan dust outbreak case (bottom).



**Table 3.** Results from evaluating the NRF simulations

| | **Russian wildfires episode** | | | | | |
| --- | --- | --- | --- | --- | --- | --- |
| | AOD | | | AE | | |
| Case | MBE | | Max. MAE | MBE | | Max. MAE |
| | Min. | Max. | | Min. | Max. | |
| ENSEMBLE | -1.55 | 0.37 | 1.58 | -0.79 | 1.09 | 1.09 |
| CS1 | -1.71 | 0.40 | 1.71 | -0.93 | 1.09 | 1.09 |
| CS2 | -1.78 | 0.16 | 1.84 | -0.96 | 0.93 | 0.98 |
| DE3 | -0.99 | 2.61 | 3.13 | | | |
| ES1 | -1.70 | 0.34 | 1.70 | -0.94 | 1 | 1 |
| ES3 | -1.82 | 0.45 | 1.82 | -0.68 | 1.24 | 1.24 |
| | **Desert dust outbreak episode** | | | | | |
| | AOD | | | AE | | |
| Case | MBE | | Max. MAE | MBE | | Max. MAE |
| | Min. | Max. | | Min. | Max. | |
| ENSEMBLE | -0.40 | 0.18 | 0.40 | -1 | 0.77 | 1.16 |
| CS1 | -0.38 | 0.19 | 0.38 | -0.79 | 1.16 | 1.16 |
| DE3 | -0.47 | 0.18 | 0.47 | | | |
| ES1 | -0.41 | 0.11 | 0.41 | -1 | 0.99 | 1.14 |
| ES3 | -0.35 | 0.40 | 0.42 | -1.43 | 0.33 | 1.43 |





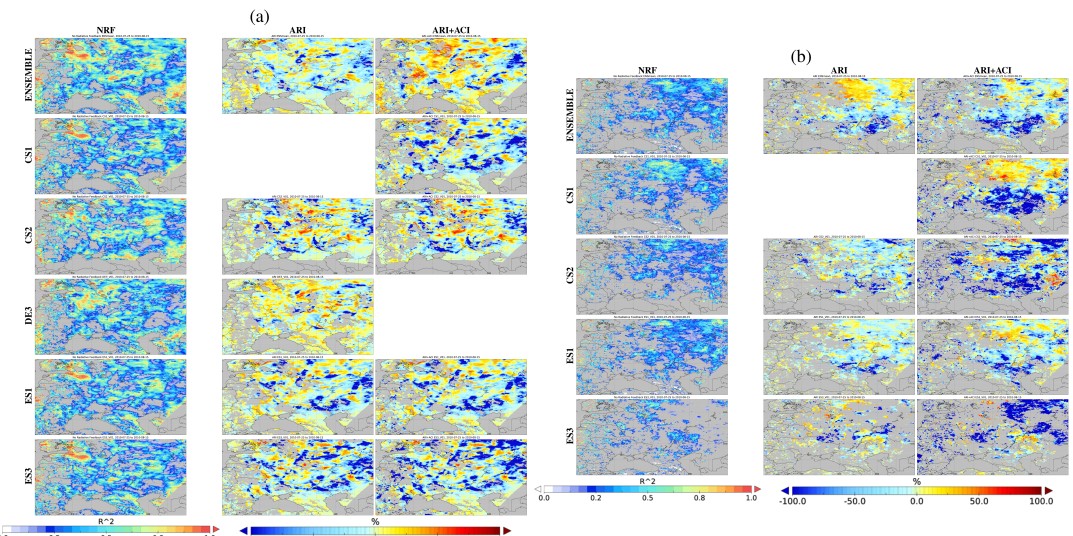

**Figure A2.** Determination coefficient for the NRF simulations (first column) and their improvements due to ARI (second) and ARI+ACI (third) of the model-MODIS comparison of AOD at $550nm$ (left) and AE between $412/470nm$ (right) for the Russian wildfires case.

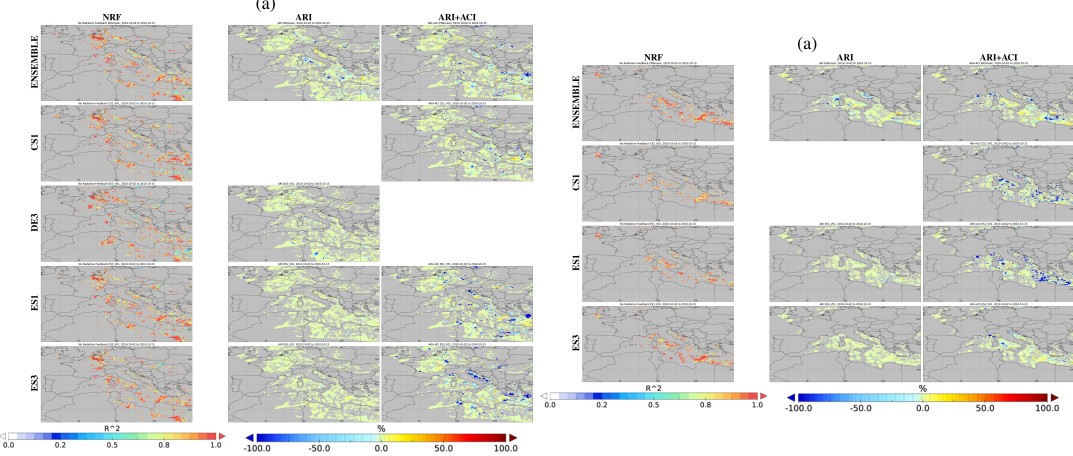

**Figure A3.** Determination coefficient for the NRF simulations (first column) and their improvements due to ARI (second) and ARI+ACI (third) of the model-MODIS comparison of AOD at $550nm$ (left) and AE between $550/860nm$ (right) for the Sahara desert dust outbreak case.