# Peer review of "An assessment of aerosol optical properties from remote sensing observations and regional chemistry-climate coupled models over Europe"

_Atmospheric Chemistry and Physics, 2017_

## Referee Comment (RC1) · Anonymous Referee #1 · 23 Oct 2017

Interactive comments on the manuscript "An assessment of aerosol optical properties from remote sensing observations and regional chemistry-climate coupled models over Europe" by Palacios-Peña et al.

General Comments

The manuscript compares aerosol optical depth and Ångström exponent retrieved from satellite platforms and simulated by distinct online coupled chemistry-meteorology models with AERONET databases during biomass burning and Saharan dust episodes
in Europe during 2010. It also analyzes if the inclusion of the aerosol-radiation and aerosol-cloud interactions improve model skills in simulating the aforementioned aerosol optical properties. The subject is of scientific relevance and within the scope of ACP. However, there are some major deficiencies, particularly concerning methodological issues and scientific arguments which must be explored in order to consider this study suitable for publication.

Specific comments

1) The authors claim to evaluate model skills comparing simulations neglecting the aerosol radiative effect with simulations performed with the aerosol direct effect and the aerosol-cloud-interactions. But in order to do so, they compare AOD and Ångström exponent fields either from AERONET or satellite retrievals. The first question is why considering or neglecting the aerosol direct effect or the aerosol-cloud interactions could improve modeling skills to reproduce AOD which depends primarily on the aerosol concentration in the atmospheric column? I would expect AOD to depend strongly on source strength (fire characteristics such as combustion phase, intensity, burnt area, injection height during the biomass burning episode and wind speed and humidity during the dust episode, for example). A better reasoning, discussing the physical mechanisms to justify how AOD field would be modified by the aerosol direct effect and the aerosol-cloud-interactions is necessary. The authors mention feedback mechanisms but a more detailed discussion on these processes should be presented. The way it was introduced is too vague (page 3, lines 10-16).

2) At page 6, line 5, it is mentioned that the heat released by the fires was not taken into account by the models. This can explain why AOD was underestimated by the models. According to Freitas et al. (2006), the heat released by the fires is responsible for the strong updrafts, transporting the emitted tracers aloft which can reach rapidly the free troposphere and even the stratosphere where they can be transported horizontally for long distances.

[Figure]

3) Was level 2.0 AERONET data used in the comparisons?

4) When comparing satellite retrievals with AERONET results, page 7, line 28, the authors mentioned that daily data were used, but none of the satellites are geostationary and aerosol optical properties can change throughout the day making no sense to compare instant values from satellite overpass time with daily mean data from AERONET. One possible methodology to follow was proposed by Ichoku et al. (2002), based on spatial mean (for satellite data) versus time mean (for AERONET data) around AERONET geographical location and satellite overpass time respectively. Since in the present study the authors have the advantage of the model results, wind speed and direction from the models can help to define the best area coverage and time interval in estimating the mean values to be compared.

5) Also concerning satellite versus AERONET results, apparently the spectral dependence was not taken into account appropriately. From Figure A1, OMI and SeaWIFS AOD at distinct channels are compared with AOD from AERONET, but AOD from AERONET was kept fixed for varying OMI/SeaWIFS wavelengths.

6) When comparing model results with either MODIS or AERONET data, it is not clear how clouds were excluded from model results, since the retrieved data either from MODIS or AERONET are available under cloud free conditions only.

7) At the Results section (page 10, lines 20-21), the physical meaning why ARI contributed to improving AOD estimation especially over the areas with high AOD values should be explored. Although the authors mentioned this improvement at regions of high AOD, I particularly cannot observe it, looking at the maps from Figure 2. High AOD areas correspond to locations with yellow color in Figure 2.a and improved results are colored in yellow to red in Figure 2.c and such areas are located in a more systematic way further to the west. In the areas of high AOD, blue and yellow/reddish colors are randomly distributed in Figure 2.c.

8) From Figure 2.a, AOD values higher than 1.5 were observed during the Russian fire

episode, but in Figures 3.b and 3.c, when comparing time series of modeling results with AERONET AOD values, the authors chose AERONET sites with AOD lower than 1.0, which we conclude are outside the smoky region. And from those graphs, when high AOD values were simulated, AERONET data did not show such enhancement, also contradicting the statement that including ARI and ARI+ACI improved modeling results.

9) In the comparison between model simulated and AERONET data for the Saharan dust episode, the time series at Figures 7.b and 7.c (page 31) and 9.b and 9.c (page 33) for the chosen AERONET sites present many days without AERONET retrievals. At Toulon, data for only 3 days were available and at Helgoland only 2. How could the results be statistically significant with so few data available to compare?

10) Page 36: The title of the table must express clearly what it shows. For example, why MBE has minimum and maximum values if, by definition, it represents a mean value?

Technical corrections

Page 2, line 27: there is a typo after "Earth";

Line 30: remove "with" between "instruments" and "onboard";

Page 3, line 8: remove "of" between "instruments" and "onboard";

Line 26: remove ";" after "demonstrate;

Line 32: replace "This" by "The";

Line 33: Please insert the correct unit for temperature in "between 0.2 to 2.6°". Note that if the unit is Kelvin (K), there is no degree symbol (°);

Page 4, lines 5-6: replace "led a" by "led to a";

Line 7: The authors mentioned a drop in the mean temperature, but it is not clear if

it refers to surface temperature or air temperature. If air temperature, please specify at each level or altitude. Moreover, as commented previously, the correct unit for temperature in Kelvin is K, without the degree symbol;

Line 20: Add a white space after "(2)";

Line 26: The correct spelling is "Ångström". Please, check throughout the entire manuscript.

Page 5, line 18-19: remove comma symbol (,) after 60° and 55°;

Line 32: remove tilde symbol ($\sim$) above 7;

Page 7, line 4: MODIS stands for Moderate Resolution Imaging Spectroradiometer;

Page 9, line 24: Figures 2 show (remove "s" from "shows");

Page 9, line 27 and page 11, line 17: remove "s" from "surroundings": surrounding areas;

Page 11, line 4: Replace "As indicated Palacios-Peña" by "As indicated by Palacios-Peña";

Line 8: Do you mean "a rough overestimation of about 50% of the emission from the total biomass burnt used here"?

Line 20: Add a white space between ")" and "indicated";

Page 12, line 3: Add a white space between "," and "the";

Line 16: replace "are" by "area";

Page 27, Figure 3: It would be helpful to identify where the sites used to generate figures (b) and (c) are located in the map of (a). This can be done using a distinct symbol for them in the maps of Figure 3(a).

References

Freitas et al. 2006, GRL 33, L17808, doi:10.1029/2006GL026608. Ichoku et al. 2002, GRL 29, doi:10.1029/2001GL013206.
* * *

---

## Referee Comment (RC2) · Anonymous Referee #2 · 31 Oct 2017

General comments: The manuscript "An assessment of aerosol optical properties from remote sensing observations and regional chemistry-climate coupled models over Europe" presents: a) An intercomparison of Aerosol Optical Depth (AOD) from distinct remote sensing platforms (ground-based and orbital) in order to identify the more accurate AOD product. Ground-based AOD retrieval from AERONET is taken as the reference to validate the remain (orbital) products.

b) Once defined the best satellite AOD products, the authors applied them, along AERONET retrieval, in the evaluation of a set of Chemistry transport models (CTMs)

simulation of Aerosol Optical Depth (AOD) and Angstrom Exponent (AE) for two aerosol events that affected Europe. One related to an episode of biomass burning in Russia and another to a Saharan dust-outbreak event. The sensitivity of the simulation of AOD and AE by distinct modelling systems to the inclusion of aerosol radiative interaction (ARI) and Aerosol Cloud Interaction(ACI) is evaluated.

The manuscript is within the scope of ACP, the issue discussed is an important topic in the field of atmospheric and climate science. The goal and the methods are clearly described, and the results has potential to contribute to the understanding and improvement of aerosol modelling capabilities over Europe. However, the paper needs some work before its publication. The discussion of the results of the study may be conducted in more concise way in order to make clearer the paper main results and it easier to the reader. Moreover, an effort to go beyond simple descriptions of what figures are showing, i.e. into a further discussion on influence of the accuracy of the representation and mechanism of aerosol effects the models analysed. I think would bring significant contribution. I would say a similar comment regarding satellite products evaluation, few is discussed regarding the essential drivers of the difference between the products investigated. I highlight these aspects taking as reference one of the goal of the manuscript, which is "...to characterize the uncertainties associated with satellite and modelling...".

Specific comments: Page 1, Line 13: "The evaluated variables were aerosol optical depth(AOD) and Angstrom Exponent (AE)..." I think it would be helpful to provide this information early in the abstract text.

Page 4, Line 9: "However, none of the aforementioned studies has evaluated the representation of aerosol optical properties and the effects of ARI+ACI on these properties." At this stage, I would suggest a short descriptive exercise on the ways through(mechanisms) which ARI and ACI can influence the simulation of AOD and AE in order to establish a theoretical reference to help the contextualization of the paper main results. For example, which impact one would expect to see on the AOD field just by adding ARI, would inclusion of ACI reinforce or counter balance ARI impact? In

other words, a short revision about what have been observed in previous study regarding the exact feedback effects that the current manuscript aims to evaluate.

Page 7, Line 12: "-0.02-10%, +0.04+10%" to "-0.02-10%*AOD, +0.04+10%*AOD". Somewhere else in the text there is a similar correction needed.

Page 8, Line 8 and 9: Define the variable correspondent to each equation (for example, NI_MAE= eq. 1)

Page 8, Line 18: "as data were not available". Which AE data is not available, from satellite or from AERONET? Clarify.

Page 8, Line 17: "Table 2" – It is not clear which AERONET wavelength the authors are comparing with the satellite wavelength, since in the manuscript is suggested 670 nm as the reference wavelength for AERONET. However, for the satellite a set of different wavelengths are presented. Similar issue occurs for Figure A1, there is no indication of wavelength in the axes, AOD at which wavelength the authors are comparing? That have to be made clear in the plots. I would extend my comments to the captions of both table and figure, needed to be auto-explicative, and it is not.

Page 9, Line 23: "Figures 2 shows the evaluation of AOD...". I would suggest the authors to think about the arrangement of the elements of Figure 2, mainly regarding the distribution of the AOD field (a). Moreover, I wonder why the authors did not considered the similar colour scale for satellite and model field, that would help.

Page 10, Line 32: "... the best skills was the Toravere station...". I consider important to show in the map the locations of the stations that the manuscript highlight as is done for Toravere, so one can have a better idea where the mode is performing better.

Page 11, Line 9 – 12: " However, our results...". I wonder about the role of emissions and meteorology (circulation/precipitation etc.) on the discrepancy between observation and models, since in the manuscript few is said on this respect. I'm a bit confuse, according to Figure A1, satellite(MODIS) seems to underestimate AOD when compared with AERONET, at least for Russian fire. Figure 3 shows that models overestimate AOD when compared with AERONET, and Figure 2 that models underestimate AOD regarding satellite. If in general models AOD is higher than AERONET and lower than satellite, how can AERONET be higher than satellite? May I have understood it wrong, but I could not figure it out.

Page 12, Line 4: "...with available AE data was very limited and substantially lower than for AOD." AERONET AE used to be in same frequency that AOD, is not that the case for your stations?

Page 12, Line 10: "3.2.2 Saharan desert dust outbreak case". I'm a bit concerned about this case, since the satellites and AERONET barely spotted this event over Europe, which clearly reflected in the amount of observational data to conduct a consistent analysis of the models simulations performance. An example, the highlighted AERONET station in Figure 7.

Page 12, Line 17: "This value was not very high for a dust outbreak, but was caused by wet deposition(rain during the episode...". This is what I was referring when claiming previously to the potential influence of meteorology on the models performance against observations, and that I think should be considered in the analysis.

Page 15, Lin 8-10: This seems to be a challenge for the manuscript discussion as whole, i.e, the separation of the impact of the issue of aerosol accurate model representation (emission/microphysics) from the impact of neglecting aerosol effects (ARI+ACI). Another point is, I recognize that ensemble may be the best options to provide a prognostic or diagnostic of an atmospheric event when one has a set of numerical simulations from distinct models. However, when the focused is to assess the effect of particular feedback mechanism, which seems to be the case, analysis should be shifted to individual model response. There are some individual analysis across the manuscript, but I think there is a particular emphasis in the ensemble results.

Technical corrections:

It seems that there is an excessive use of bracket across the text, the authors may re-evaluate when it is necessary to use this resource.

In general, most of captions of Figures and Tables need more details description.

Page 1, Line 2: "...uncertain forcing agents..." to "... uncertain climate forcing agents ..."

Page 1, Line 5: "... inclusion of aerosol-radiation (ARI) or aerosol-cloud interactions (ACI) helps improve..." to "... inclusion of aerosol-radiation (ARI) or/and aerosol-cloud interactions (ACI) helps to improve..."

Page 1, Line 8: "... Mediterranean Sea..." to "... the Mediterranean Sea..."

Page 2, Line 21: "The main advantages of remote sensing are: (1) they do not perturb the observed..." to "The main advantages of remote sensing are: (1) it do not perturb the observed..."

Page 2, Line 32: "There are instruments with onboard satellites..." to "There are instruments aboard satellites..."

Page 3, Line 8: "...different instruments of onboard satellites..." to "...different instruments aboard of satellites..."

Page 4, Line 8-9: "...studies has evaluated..." to "...studies have evaluated..."

Page 6, Line 24: "...Forkel et al(2015), Im et al..." to "...Forkel et al(2015) and Im et al..."

Page 6, Line 25: "...Chapman et al (2009), Barnard et al..." to "...Chapman et al (2009) and Barnard et al.."

Page 7, Line 4: "...on board a satellites..." to "...aboard satellites,..."

Page 11, Line 5: "...as indicated Palacios-Penã..." to "...as indicated in Palacios-Penã..."

Page 13, Line 6: "...at the PM10 levels..." to "...of the PM10 levels..."

---

## Author Comment (AC1) · 26 Dec 2017

A: First, we would like to thank the anonymous referees for their valuable comments in the interactive comment on "An assessment of aerosol optical properties from remote sensing observations and regional chemistry-climate coupled models over Europe" by Laura Palacios-Peña et al. The manuscript has been revised after the reviewer's comments in order to correct errors and to introduce the reviewers' suggestions for improving the quality of the paper. Please see below our point-by-point replies:

*Anonymous Referee #1*

General Comments

The manuscript compares aerosol optical depth and Ångström exponent retrieved from satellite platforms and simulated by distinct online coupled chemistry-meteorology models with AERONET databases during biomass burning and Saharan dust episodes in Europe during 2010. It also analyzes if the inclusion of the aerosol-radiation and aerosol-cloud interactions improve model skills in simulating the aforementioned aerosol optical properties. The subject is of scientific relevance and within the scope of ACP. However, there are some major deficiencies, particularly concerning methodological issues and scientific arguments which must be explored in order to consider this study suitable for publication.

Specific comments

1) The authors claim to evaluate model skills comparing simulations neglecting the aerosol radiative effect with simulations performed with the aerosol direct effect and the aerosol-cloud-interactions. But in order to do so, they compare AOD and Ångström exponent fields either from AERONET or satellite retrievals. The first question is why considering or neglecting the aerosol direct effect or the aerosol-cloud interactions could improve modeling skills to reproduce AOD which depends primarily on the aerosol concentration in the atmospheric column? I would expect AOD to depend strongly on source strength (fire characteristics such as combustion phase, intensity, burnt area, injection height during the biomass burning episode and wind speed and humidity during the dust episode, for example). A better reasoning, discussing the physical mechanisms to justify how AOD field would be modified by the aerosol direct effect and the aerosol-cloud-interactions is necessary. The authors mention feedback mechanisms but a more detailed discussion on these processes should be presented. The way it was introduced is too vague (page 3, lines 10-16).

A: As both reviewers pointed out the necessity to shed some light on the physical mechanisms driving aerosol-radiation (ARI) and aerosol-clouds interactions (ACI) to modify AOD. A review of the impacts of these interactions was included in pages 3 and 4 of the original manuscript but focused on meteorological forecasts. However, this section has been re-written in order to include the impacts on AOD fields.

*"However, none of the aforementioned studies have evaluated the representation of aerosol optical properties and the effects of ARI+ACI on these properties. As the previous revision reveals, a strong influence of the ARI+ACI exists on meteorological variables. Several studied untangled the influence of atmospheric aerosols on the atmospheric boundary layer (Roy and Sharp,2013), atmospheric stability and winds (Péré et al., 2014; Baró et al., 2017) with a consequent effect on aerosol concentrations (Zhang et al., 2015). The winds variation affects emissions of wind-dependent particles over land, such as desert dust, or ocean, sea salt (Boucher et al., 2013; Prijith et al., 2014; Li et al., 2015). Moreover, the transport of pollutants (e.g. Yang et al., 2017) and the aerosol vertical distribution (Mishra et al., 2015) could be altered. The modification of all of these process influences on aerosol optical properties (e.g. Huang et al., 2010). On the other hand, ARI+ACI effects on relative humidity*

*(Baró et al., 2017) modify the size of the particles due to the hygroscopic growth affecting the particle extinction (Curci et al., 2015)."*

Baró, R., Lorente-Plazas, R., Montávez, J., and Jiménez-Guerrero, P.: Biomass burning aerosol impact on surface winds during the 2010 Russian heat wave, Geophysical Research Letters, 44, 1088–1094, 2017.

Boucher, O., Randall, D., Artaxo, P., Bretherton, C., Feingold, G., Forster, P., Kerminen, V.-M., Kondo, Y., Liao, H., Lohmann, U., Rasch, P., Satheesh, S., Sherwood, S., Stevens, B., and Zhang, X.: Clouds and aerosols, in: Climate change 2013: The physical science basis. Contribution of Working Group I to the Fourth Assessment Report of the International Panel of Climate Change, pp. 571–657, Cambridge University Press, 2013.

Curci, G., Hogrefe, C., Bianconi, R., Im, U., Balzarini, A., Baró, R., Brunner, D., Forkel, R., Giordano, L., Hirtl, M., et al.: Uncertainties of simulated aerosol optical properties induced by assumptions on aerosol physical and chemical properties: An AQMEII-2 perspective, Atmospheric Environment, 115, 541–552, 2015.

Huang, H., Thomas, G., and Grainger, R.: Relationship between wind speed and aerosol optical depth over remote ocean, Atmospheric Chemistry and Physics, 10, 5943–5950, 2010.

Li, S., Wang, T., Xie, M., Han, Y., and Zhuang, B.: Observed aerosol optical depth and Ångström exponent in urban area of Nanjing, China, Atmospheric Environment, 123, 350–356, 2015.

Mishra, A. K., Koren, I., and Rudich, Y.: Effect of aerosol vertical distribution on aerosol-radiation interaction: A theoretical prospect, Heliyon, 1, e00 036, 2015.

Péré, J., Bessagnet, B., Mallet, M., Waquet, F., Chiapello, I., Minvielle, F., Pont, V., and Menut, L.: Direct radiative effect of the Russian wildfires and its impact on air temperature and atmospheric dynamics during August 2010, Atmospheric Chemistry and Physics, 14, 1999–2013, 2014.

Prijith, S., Aloysius, M., and Mohan, M.: Relationship between wind speed and sea salt aerosol production: A new approach, Journal of Atmospheric and Solar-Terrestrial Physics, 108, 34–40, 2014.

Roy, S. and Sharp, J.: Why Atmospheric Stability Matters in Wind Assessment, North American Wind Power Available at: <http://nawindpower.com/online/issues/NAW1301/FEAT_06_Why_Atmospheric.html> [Accessed April 2016], 2013.

Yang, Y., Russell, L. M., Lou, S., Liao, H., Guo, J., Liu, Y., Singh, B., and Ghan, S. J.: Dust-wind interactions can intensify aerosol pollution over eastern China, Nature Communications, 8, 2017.

Zhang, B., Wang, Y., and Hao, J.: Simulating aerosol–radiation–cloud feedbacks on meteorology and air quality over eastern China under severe haze conditions in winter, Atmospheric Chemistry and Physics, 15, 2387–2404, 2015.

2) At page 6, line 5, it is mentioned that the heat released by the fires was not taken into account by the models. This can explain why AOD was underestimated by the models. According to Freitas et al. (2006), the heat released by the fires is responsible for the strong updrafts, transporting the emitted tracers aloft which can reach rapidly the free troposphere and even the stratosphere where they can be transported horizontally for long distances.

A: The reviewer raises a very interesting point and this explanation has been included in the conclusion sections as a possible cause of the underestimation of AOD due to the understated injection height of the total biomass burning emissions.

Page 11:*"On the other hand the heat released by the fires, which is responsible for the strong updrafts transporting the emitted tracers rapidly to the free troposphere and even the stratosphere (Freitas et al. 2007), was not taken into account in the simulations and could affect the aerosol vertical distribution."*

Page 14: *"AOD was generally underestimated by all simulations. This behaviour could be explained by a misunderstanding in the aerosol vertical distribution due to the understated injection height of the total biomass burning emissions or because the heat released by the fires was not taken into account, which could affect the representation of AOD. "*

3) Was level 2.0 AERONET data used in the comparisons?

A: The reviewer is right and the level 2.0 AERONET data was used in the comparisons. This information has been included in the manuscript.

*Page 7. Line 6-8. "The data used from AERONET were AOD of Level 2.0 at different wavelengths from the available European stations during these episodes. The variables used to the satellite-AERONET comparison were AOD at the closer wavelength to satellite retrievals and those used to the models-AERONET comparison were AOD 670nm and AE between 440/870nm....".*

4) When comparing satellite retrievals with AERONET results, page 7, line 28, the authors mentioned that daily data were used, but none of the satellites are geostationary and aerosol optical properties can change throughout the day making no sense to compare instant values from satellite overpass time with daily mean data from AERONET. One possible methodology to follow was proposed by Ichoku et al. (2002), based on spatial mean (for satellite data) versus time mean (for AERONET data) around AERONET geographical location and satellite overpass time respectively. Since in the present study the authors have the advantage of the model results, wind speed and direction from the models can help to define the best area coverage and time interval in estimating the mean values to be compared.

A: It is possible that the text of our manuscript could be misleading. We did not compare instant values from satellite with daily mean data from AERONET. We compared daily means from satellite with the daily mean from AERONET. Although the reviewer is right that these satellites are not geostationary the daily means were computed with an enough number of observations in order to become the results reliable. Moreover, the data used to compute this daily mean presented the highest confidence flag value. For example, OMI Level 3 daily global products are produced by 14 files of data from single orbit per day (OMI Team, 2012).

OMI Team: Ozone Monitoring Instrument (OMI)Data User's Guide, 2012.

5) Also concerning satellite versus AERONET results, apparently the spectral dependence was not taken into account appropriately. From Figure A1, OMI and SeaWIFS AOD at distinct channels are compared with AOD from AERONET, but AOD from AERONET was kept fixed for varying OMI/SeaWIFS wavelengths.

A: The OMI and SeaWIFS AOD at distinct channels were evaluated against AOD from AERONET at the same wavelength. We have indicated this information in the "Observational data" section as we mentioned above in the specific comment 3).

6) When comparing model results with either MODIS or AERONET data, it is not clear how clouds were excluded from model results, since the retrieved data either from MODIS or AERONET are available under cloud free conditions only.

A: As the reviewer reveals, it is not clearly explained how the clouds were excluded. When the models were compared with the satellite data, only those data with matched in time and space with the satellite retrieval was selected (that is, only under cloud free conditions). We have modified the text in page 8, line 3-7 in order to clarify this issue.

*"Satellite data and model data were bilinearly interpolated to a common working grid, which corresponded with the analysis grid (described above) according to case studies. After the interpolation, and in order to compare with AERONET, the satellite and simulations data were extracted from the cell that covered the corresponding station coordinates of the AERONET station by a nearest neighbour approach. When the models were compared with the satellite data, only those data with matched in time and space with the satellite retrievals were selected. The evaluation was done using both available the daily and hourly data."*

7) At the Results section (page 10, lines 20-21), the physical meaning why ARI contributed to improving AOD estimation especially over the areas with high AOD values should be explored. Although the authors mentioned this improvement at regions of high AOD, I particularly cannot observe it, looking at the maps from Figure 2. High AOD areas correspond to locations with yellow color in Figure 2.a and improved results are colored in yellow to red in Figure 2.c and such areas are located in a more systematic way further to the west. In the areas of high AOD, blue and yellow/reddish colors are randomly distributed in Figure 2.c.

A: The reviewer is right and the Results section has been changed in order to get a better description of the results.

8) From Figure 2.a, AOD values higher than 1.5 were observed during the Russian fire episode, but in Figures 3.b and 3.c, when comparing time series of modeling results with AERONET AOD values, the authors chose AERONET sites with AOD lower than 1.0, which we conclude are outside the smoky region. And from those graphs, when high AOD values were simulated, AERONET data did not show such enhancement, also contradicting the statement that including ARI and ARI+ACI improved modeling results.

A: As we indicated in the manuscript, we stablished a requirement of significance for the AERONET results, which consisted in showing only those stations where correlation values were significant at the 95% level. Following this criterion, Figure 3.b and 3.c displays the behaviour of the station with the highest significant correlations values regardless where they are. As it was explained in the manuscript (page 11, line 1-8), *"The station where the NRF, ARI and ARI+ACI ensemble means showed the best skills was the Toravere station (correlation coefficient for NRF of 0.68, ARI 0.73 and ARI+ACI 0.73, Figure 3b). At this station (located in northerly areas) AOD was higher between 25 and 30 July, and between 5 and 10 of August. However, the maximum correlation values among all the experiments were found at the Efoire station (Figure 3c) for the CS2 configuration (correlation coefficient for NRF 0.83, ARI 0.84 and ARI+ACI 0.82). This can be explained by the enhanced resolution of CS2 around 9.9km. This fine resolution may lead to improvements in the local representation of AOD. It should be highlighted that no clear improvement in the model-AERONET correlations was noted when the aerosol radiative feedbacks were taken into account."*

9) In the comparison between model simulated and AERONET data for the Saharan dust episode, the time series at Figures 7.b and 7.c (page 31) and 9.b and 9.c (page 33) for the chosen AERONET sites present many days without AERONET retrievals. At Toulon, data for only 3 days were available and at Helgoland only 2. How could the results be statistically significant with so few data available to compare?

A: As we indicated in the previous point, we stablished a requirement of significance for the AERONET results. Thus, although these stations show only 3 and 2 days of data, we considered the results statistically significant.

10) Page 36: The title of the table must express clearly what it shows. For example, why MBE has minimum and maximum values if, by definition, it represents a mean value?

A: The reviewer is right. We clarified the values represented in the table. When we indicated minimum and maximum values we were talking about a spatial distribution of the MBE or MAE. This has been clarified in the title of the table.

*"Table 3. Spatial results from evaluating the NRF simulations"*

Technical corrections

A: Please find below the list of recommendations of the Reviewer and the corrections made.

Page 2, line 27: there is a typo after "Earth"; Line 30: remove "with" between "instruments" and "onboard"; (Done)

Page 3, line 8: remove "of" between "instruments" and "onboard"; (Done)

Line 26: remove ";" after "demonstrate; (Done)

Line 32: replace "This" by "The"; (Done)

Line 33: Please insert the correct unit for temperature in "between 0.2 to 2.6º". Note that if the unit is Kelvin (K), there is no degree symbol (º); (Done)

Page 4, lines 5-6: replace "led a" by "led to a"; (Done)

Line 7: The authors mentioned a drop in the mean temperature, but it is not clear if it refers to surface temperature or air temperature. If air temperature, please specify at each level or altitude. Moreover, as commented previously, the correct unit for temperature in Kelvin is K, without the degree symbol; (Done)

Line 20: Add a white space after "(2)"; (Done)

Line 26: The correct spelling is "Ångström". Please, check throughout the entire manuscript. (Done)

Page 5, line 18-19: remove comma symbol (,) after 60º and 55º; (Done)

Line 32: remove tilde symbol (~) above 7; (Done)

Page 7, line 4: MODIS stands for Moderate Resolution Imaging Spectroradiometer; (Done)

Page 9, line 24: Figures 2 show (remove "s" from "shows"); (Done)

Page 9, line 27 and page 11, line 17: remove "s" from "surroundings": surrounding areas; (Done)

Page 11, line 4: Replace "As indicated Palacios-Peña" by "As indicated by Palacios-Peña"; (Done)

> Line 8: Do you mean "a rough overestimation of about 50% of the emission from the total biomass burnt used here"?
>
> A: We meant "a rough overestimation at about 50% of the total biomass burning emissions used here". There was a typo which has corrected. (Done)
>
> Line 20: Add a white space between ")" and "indicated"; (Done)

Page 12, line 3: Add a white space between "," and "the"; Line 16: replace "are" by "area"; (Done)

Page 27, Figure 3: It would be helpful to identify where the sites used to generate figures (b) and (c) are located in the map of (a). This can be done using a distinct symbol for them in the maps of Figure 3(a). (Done)

Freitas et al. 2006, GRL 33, L17808, doi:10.1029/2006GL026608.

Ichoku et al. 2002, GRL 29, doi:10.1029/2001GL013206.

**Anonymous Referee #2**

General comments

The manuscript "An assessment of aerosol optical properties from remote sensing observations and regional chemistry-climate coupled models over Europe" presents:

a) An intercomparison of Aerosol Optical Depth (AOD) from distinct remote sensing platforms (ground-based and orbital) in order to identify the more accurate AOD product. Ground-based AOD retrieval from AERONET is taken as the reference to validate the remain (orbital) products.

b) Once defined the best satellite AOD products, the authors applied them, along AERONET retrieval, in the evaluation of a set of Chemistry transport models (CTMs) simulation of Aerosol Optical Depth (AOD) and Ångström Exponent (AE) for two aerosol events that affected Europe. One related to an episode of biomass burning in Russia and another to a Saharan dust-outbreak event. The sensitivity of the simulation of AOD and AE by distinct modelling systems to the inclusion of aerosol radiative interaction (ARI) and Aerosol Cloud Interaction(ACI) is evaluated.

The manuscript is within the scope of ACP, the issue discussed is an important topic in the field of

atmospheric and climate science. The goal and the methods are clearly described, and the results has potential to contribute to the understanding and improvement of aerosol modelling capabilities over Europe. However, the paper needs some work before its publication. The discussion of the results of the study may be conducted in more concise way in order to make clearer the paper main results and it easier to the reader. Moreover, an effort to go beyond simple descriptions of what figures are showing, i.e. into a further discussion on influence of the accuracy of the representation and mechanism of aerosol effects the models analysed. I think would bring significant contribution. I would say a similar comment regarding satellite products evaluation, few is discussed regarding the essential drivers of the difference between the products investigated. I highlight these aspects taking as reference one of the goal of the manuscript, which is ". . .to characterize the uncertainties associated with satellite and modelling...".

A: We thank the Anonymous Referee 2 for his/her valuable comments. An effort has been made in order to include the above reviewer's suggestions in the manuscript.

Specific comments

Page 1, Line 13: "The evaluated variables were aerosol optical depth (AOD) and Ångström Exponent (AE)..." I think it would be helpful to provide this information early in the abstract text.

A: The first paragraph of the abstract has been modified in order to include the reviewer suggestion.

*"Atmospheric aerosols modify the radiative budget of the Earth due to their optical, microphysical and chemical properties, and are considered one of the most uncertain climate forcing agents. In order to characterise the uncertainties associated with satellite and modelling approaches to represent aerosol optical properties, mainly aerosol optical depth (AOD) and the Ångström exponent (AE), their representation by different remote sensing sensors and regional online coupled chemistry-climate models over Europe is evaluated. This work also characterizes whether the inclusion of aerosol-radiation (ARI) or/and aerosol-cloud interactions (ACI) helps improve the skills of modelling outputs."*

Page 4, Line 9: "However, none of the aforementioned studies has evaluated the representation of aerosol optical properties and the effects of ARI+ACI on these properties." At this stage, I would suggest a short descriptive exercise on the ways through (mechanisms) which ARI and ACI can influence the simulation of AOD and AE in order to establish a theoretical reference to help the contextualization of the paper main results. For example, which impact one would expect to see on the AOD field just by adding ARI, would inclusion of ACI reinforce or counter balance ARI impact? In other words, a short revision about what have been observed in previous study regarding the exact feedback effects that the current manuscript aims to evaluate.

A: As explained, we have re-written part of the Introduction section in order to clarify the physical mechanisms through aerosol-radiation (ARI) and aerosol-clouds interactions (ACI) modified AOD field as both reviewers suggested.

Page 7, Line 12: "-0.02-10%, +0.04+10%" to "-0.02-10%*AOD, +0.04+10%*AOD". Somewhere else in the text there is a similar correction needed.

A: This correction has been done and the text has been reviewed in order to correct similar expressions.

Page 8, Line 8 and 9: Define the variable correspondent to each equation (for example, NI_MAE= eq. 1).

A: The variables indicated by the reviewer have been re-defined. The Normalized Improvement of the MAE has been defined as NI_MAE = Eq.1 and the Normalized Improvement of the Coefficient of Determination as NI_$R^2$ = Eq.2.

Page 8, Line 18: "as data were not available". Which AE data is not available, from satellite or from AERONET? Clarify.

A: This sentence has been modified in order to clarify the information. *"AE was excluded from this analysis as satellite data matching with AERONET site were not enough, which made the evaluation statistically non-significant."*

Page 8, Line 17: "Table 2" – It is not clear which AERONET wavelength the authors are comparing with the satellite wavelength, since in the manuscript is suggested 670 nm as the reference wavelength for AERONET. However, for the satellite a set of different wavelengths are presented. Similar issue occurs for Figure A1, there is no indication of wavelength in the axes, AOD at which wavelength the authors are comparing? That have to be made clear in the plots. I would extend my comments to the captions of both table and figure, needed to be auto-explicative, and it is not.

A: The information about the AERONET wavelength used was confused in the manuscript. For this reason and following the suggestions of both reviewers we expanded this information in Page 7. Line 6-8. *"The data used from AERONET were AOD of Level 2.0 at different wavelengths from the available European stations during these episodes. The variables used to the satellite-AERONET comparison were AOD at the closer wavelength to satellite retrievals and those used to the models-AERONET comparison were AOD 670nm and AE between 440/870nm...."*.

Moreover, the captions of both table and figure have been redefined and the quality of Figure A1 has improved.

*"Table 2. Coefficient of correlation of the comparison between satellites and the AERONET AOD data."*

*"Figure A1. Satellite-AERONET linear regression. Russian wildfires case (top) and Saharan dust outbreak case (bottom). AOD from MOD04_L2 and MYD04_L2 at 550nm. For OMI and Sea WIFS AOD at indicated wavelength. "*

Page 9, Line 23: "Figures 2 shows the evaluation of AOD...". I would suggest the authors to think about the arrangement of the elements of Figure 2, mainly regarding the distribution of the AOD field (a). Moreover, I wonder why the authors did not considered the similar colour scale for satellite and model field, that would help.

A: The set of images shown in the Figure 2 depict the results for the evaluation of AOD at 550nm. But we modified the text in order to clarify this sentence. "Figure 2 shows the evaluation of AOD at 550nm and Figure 4 of AE between 412nm and 470nm. From the AERONET data, Figure 3 displays the results for AOD at 670nm and Figure 5 for AE between 440/870nm"

On the other hand, we use different colour scale because we represent different results as indicated by the caption of the figure. *"Model-MODIS comparison of AOD at 550nm for the Russian wildfires case: (a) Satellite values; (b) MBE for NRF (first column), ARI (second) and ARI+ACI (third); and (c) MAE for the NRF simulations (first column) and their improvements due to ARI (second) and ARI+ACI (third)."* Thus, Figure 2a shows the satellite fields while figure 2b shows the MBE results and figure 2c the MAE results and their improvements.

Page 10, Line 32: ". . . the best skills was the Toravere station...". I consider important to show in the map the locations of the stations that the manuscript highlight as is done for Toravere, so one can have a better idea where the mode is performing better.

A: Following both reviewers' suggestions, we modified figures 3,5,7 and 9 to point out where the location of the stations that the manuscript highlight is.

Page 11, Line 9 – 12: "However, our results...". I wonder about the role of emissions and meteorology

(circulation/precipitation etc.) on the discrepancy between observation and models, since in the manuscript few is said on this respect. I'm a bit confuse, according to Figure A1, satellite (MODIS) seems to underestimate AOD when compared with AERONET, at least for Russian fire. Figure 3 shows that models overestimate AOD when compared with AERONET, and Figure 2 that models underestimate AOD regarding satellite. If in general models AOD is higher than AERONET and lower than satellite, how can AERONET be higher than satellite? May I have understood it wrong, but I could not figure it out.

A: On the one hand, we have tried to extend in the text the role of emissions and meteorology on the discrepancy between observation and models. On the other hand, and regarding AOD in Figures A1, 2 and 3, the stations chosen in Figure 3 might lead to a misinterpretation of the results. As the reviewers suggested, we modified figures 3,5,7 and 9 to point out where the location of the stations is. Considering the location of the Toravere and Efoire stations (represented in Figure 3) and comparing with Figure 2, it can be found that the areas close to these stations presented very low underestimations and even for some models an overestimation of AOD.

Page 12, Line 4: "...with available AE data was very limited and substantially lower than for AOD." AERONET AE used to be in same frequency that AOD, is not that the case for your stations?

A: This part of the text was misunderstanding and for this reason, has re-written. It is not the number of station with available data but the number of station with significant results.

*"...the number of stations with significant results for AE data was very limited and substantially lower than for AOD and the results show lower correlation coefficient values. "*

Page 12, Line 10: "3.2.2 Saharan desert dust outbreak case". I'm a bit concerned about this case, since the satellites and AERONET barely spotted this event over Europe, which clearly reflected in the amount of observational data to conduct a consistent analysis of the models' simulations performance. An example, the highlighted AERONET station in Figure 7.

A: We fully agree with the reviewer's comment. The amount of observation data is low but one of the objectives of this work was to evaluate the aerosol optical representation within the ES1004 COST Action EuMetChem. As indicated in the manuscript, the working group 2 of this action investigated the importance of different processes and feedback in online coupled chemistry-meteorology models for air quality simulations and weather forecasts. For this reason, two different episodes with a high aerosol load over Europe in 2010 were chosen from the previous experience in AQMEII phase 2, where the entire 2010 year was simulated.

In spite of the low amount of observational data, the results of the comparison of simulations versus AERONET showed a significance at 95% level.

Page 12, Line 17: "This value was not very high for a dust outbreak, but was caused by wet deposition (rain during the episode...". This is what I was referring when claiming previously to the potential influence of meteorology on the models' performance against observations, and that I think should be considered in the analysis.

A: Following the reviewer's suggestions, we have taken into account the potential influence of meteorology when we have re-written the analysis.

Page 15, Lin 8-10: This seems to be a challenge for the manuscript discussion as whole, i.e, the separation of the impact of the issue of aerosol accurate model representation (emission/microphysics) from the impact of neglecting aerosol effects (ARI+ACI). Another point is, I recognize that ensemble may be the best options to provide a prognostic or diagnostic of an atmospheric event when one has a set of numerical simulations from distinct models. However, when the focused is to assess the effect of

particular feedback mechanism, which seems to be the case, analysis should be shifted to individual model response. There are some individual analysis across the manuscript, but I think there is a particular emphasis in the ensemble results.

A: We have clarified the wording of the page 15, line 8-10. We tried to separate the impact of the issue of aerosol accurate model representation from the impact of neglecting aerosol effects (ARI+ACI). In this precise case, the sentence has been modified in order to explain better the meaning of the conclusion: "However, we can find a better improvement due to the inclusion of ARI+ACI over the Mediterranean Sea with values around 75\%, which can be explained by a better representation of the coarse mode in the sectional approach when the radiative feedbacks were taking into account". Moreover, the umbrella of EuMetChem Cost Action led us to focus on the analysis of the ensemble results. It is of noticeable importance how the ensemble results can strongly improve the skills of individual models and therefore points out to the necessity of building ensemble of simulations when reproducing the impacts of aerosols.

Technical corrections

A: Please find below the list of recommendations of the Reviewer and the corrections made.

It seems that there is an excessive use of bracket across the text, the authors may re-evaluate when it is necessary to use this resource.

In general, most of captions of Figures and Tables need more details description.

A: We have included all the reviewer's suggestions.

Page 1, Line 2: "...uncertain forcing agents..." to "... uncertain climate forcing agents..." (Done)

Page 1, Line 5: "... inclusion of aerosol-radiation (ARI) or aerosol-cloud interactions (ACI) helps improve..." to "... inclusion of aerosol-radiation (ARI) or/and aerosol-cloud interactions (ACI) helps to improve..." (Done)

Page 1, Line 8: "... Mediterranean Sea..." to "... the Mediterranean Sea. . ." (Done)

Page 2, Line 21: "The main advantages of remote sensing are: (1) they do not perturb the observed..." to "The main advantages of remote sensing are: (1) it does not perturb the observed..." (Done)

Page 2, Line 32: "There are instruments with onboard satellites..." to "There are instruments aboard satellites..." (Done)

Page 3, Line 8: "...different instruments of onboard satellites..." to "...different instruments aboard of satellites..." (Done)

Page 4, Line 8-9: "...studies has evaluated..." to "...studies have evaluated..." (Done)

Page 6, Line 24: "...Forkel et al (2015), Im et al..." to "...Forkel et al (2015) and Im et al..." (Done)

Page 6, Line 25: "...Chapman et al (2009), Barnard et al..." to "...Chapman et al (2009) and Barnard et al." (Done)

Page 7, Line 4: "...on board satellites..." to "...aboard satellites..." (Done)

Page 11, Line 5: "...as indicated Palacios-Peña..." to "...as indicated in Palacios-Peña..." (Done)

Page 13, Line 6: "...at the PM10 levels..." to "...of the PM10 levels..." (Done)

---

## Author Comment (AC2) · 26 Dec 2017

The comment was uploaded in the form of a supplement:
https://www.atmos-chem-phys-discuss.net/acp-2017-877/acp-2017-877-AC2-supplement.pdf

---

## Author Response (AR2)

[revised manuscript text omitted]

| Aerosol option | MADE-SORGAM (??) | MADE-SORGAM (??) | ? | MADE-SORGAM (??) | MOSAIC (?) |
| Microphysic option | ? | ? | Kessler-type bulk (?) | ? | ? |
| Wet deposition option | Simple | ? | (?) | ? | Simple |
| Wet | Simple | ? | | ? | Simple |

**Table 2.** Coefficient of correlation of the comparison between satellites and the AERONET AOD data

| Case | MOD04_L2 | | MYD04_L2 | | OMI | | SeaWIFS | | | |
|---|---|---|---|---|---|---|---|---|---|---|
| Russian wildfires | Deep Blue | 0.82 | Deep Blue | 0.80 | $342.5nm$ | 0.80 | | | | |
| | Combined | 0.66 | Combined | 0.70 | $388nm$ | 0.76 | | | | |
| | Dark Target | 0.76 | Dark Target | 0.79 | $442nm$ | 0.75 | | | | |
| Dust outbreak | Deep Blue | 0.45 | Deep Blue | 0.64 | $342.5nm$ | 0.26 | $412nm\_L$ | 0.69 | $550nm$ | 0.74 |
| | Combined | 0.77 | Combined | 0.92 | $388nm$ | 0.26 | $490nm\_L$ | 0.69 | $670nm\_O$ | 0.90 |
| | | | | | | | | | $670nm\_L$ | 0.53 |
| | Dark Target | 0.77 | Dark Target | 0.92 | $442nm$ | 0.26 | $510nm\_O$ | 0.89 | $865nm\_O$ | 0.90 |

*L*: satellite product over land, *O*: satellite product over ocean

**Table 3.** Spatial results from evaluating the NRF simulations

| | **Russian wildfires episode** | | | | | |
|---|---|---|---|---|---|---|
| | AOD | | | AE | | |
| Case | MBE | | Max. MAE | MBE | | Max. MAE |
| | Min. | Max. | | Min. | Max. | |
| ENSEMBLE | -1.55 | 0.37 | 1.58 | -0.79 | 1.09 | 1.09 |
| CS1 | -1.71 | 0.40 | 1.71 | -0.93 | 1.09 | 1.09 |
| CS2 | -1.78 | 0.16 | 1.84 | -0.96 | 0.93 | 0.98 |
| DE3 | -0.99 | 2.61 | 3.13 | | | |
| ES1 | -1.70 | 0.34 | 1.70 | -0.94 | 1 | 1 |
| ES3 | -1.82 | 0.45 | 1.82 | -0.68 | 1.24 | 1.24 |

| | **Desert dust outbreak episode** | | | | | |
|---|---|---|---|---|---|---|
| | AOD | | | AE | | |
| Case | MBE | | Max. MAE | MBE | | Max. MAE |
| | Min. | Max. | | Min. | Max. | |
| ENSEMBLE | -0.40 | 0.18 | 0.40 | -1 | 0.77 | 1.16 |
| CS1 | -0.38 | 0.19 | 0.38 | -0.79 | 1.16 | 1.16 |
| DE3 | -0.47 | 0.18 | 0.47 | | | |
| ES1 | -0.41 | 0.11 | 0.41 | -1 | 0.99 | 1.14 |
| ES3 | -0.35 | 0.40 | 0.42 | -1.43 | 0.33 | 1.43 |